# Attention Trajectories as a Diagnostic Axis for Deep Reinforcement Learning

## Abstract

The emergence and evolution of feature reliance in deep reinforcement learning agents remain poorly understood. Here, we introduce a methodological framework for analyzing the learning process through quantitative analysis of saliency maps. This approach aggregates saliency information at the object and modality level into hierarchical attention profiles, quantifying how agents allocate attention over time, thereby forming attention trajectories throughout training. These profiles are then compared across controlled conditions, connected to behavioral measurements and reproduced with different saliency methods to assess the robustness of the findings. Applied to Atari 2600 benchmarks, custom Pong environments, and biomechanical user simulations in visuomotor tasks, this framework uncovers algorithm-specific attention biases, diagnosed unintended reward-driven strategies, and overfitting to redundant sensory channels. These patterns correspond to measurable behavioral differences, demonstrating empirical links between attention profiles, learning dynamics, and agent behavior. The results establish attention trajectories as a promising diagnostic axis for tracing how feature reliance develops during training and for identifying biases and vulnerabilities invisible to performance metrics alone.

## 1 Introduction

Deep reinforcement learning (DRL) has achieved remarkable success in domains ranging from robotics to simulated users to healthcare (Han et al., 2023; Liu et al., 2022; Jayaraman et al., 2024; Fischer et al., 2024; Selder et al., 2025). Despite these advances, the learning process of DRL agents remains poorly understood. In particular, it is unclear how agents learn to rely on specific features when making decisions, how these dependencies evolve during training, and what factors influence this process. This opacity hinders not only scientific progress for improving how DRL agents learn, but also the safe and reliable deployment of DRL in real-world settings.

Among existing interpretability tools, saliency maps are used to highlight the input features most relevant for a model's prediction (Zeiler & Fergus, 2014; Selvaraju et al., 2017; Huber et al., 2022). Their post-hoc nature allows them to be applied to diverse architectures and problem settings, offering a versatile means of analyzing deep neural networks. In DRL, they have been used to visualize action-relevant regions and to expose issues such as observational overfitting (Puri et al., 2019; Song et al., 2019). However, their potential to study the learning process of DRL agents and extract global patterns of attention remains largely underexplored. The majority of prior work focuses on developing new saliency methods Greydanus et al. (2018); Iyer et al. (2018); Puri et al. (2019); Lapuschkin et al. (2019); Huber et al. (2021), while only a few studies apply existing techniques to analyze DRL agents Markovikj (2022); Wang et al. (2016); Yang et al. (2018), and even fewer attempt to extract global patterns of attention Guo et al. (2021); Lapuschkin et al. (2019), that is to say, consistent, system-level regularities in how attention is allocated across agents over time. Indeed, saliency maps are typically applied to a few isolated frames from already trained agents followed by a human interpretation of the agent's strategy. While such analyses are valuable for improving saliency techniques and providing a glimpse of what features DRL agents rely on, they do not extract information about global patterns of attention and provide very limited insight into how feature dependencies change during training. Furthermore, this qualitative and case-based usage is prone to post-hoc storytelling and confirmation bias,

a limitation highlighted by Atrey et al. (2019), who found that only 7% of interpretability claims were experimentally verified.

Here, we propose a methodological framework that uses existing saliency tools to extract global patterns of attention trajectory in DRL agents and relate them to behavior through controlled experiments. More specifically, our approach analyzes how an agent's attention evolves throughout training and identifies the factors that shape this evolution. To reduce the risks of post-hoc story telling and cognitive bias commonly associated with saliency maps, this framework is composed of five components: (i) quantification of saliency maps, (ii) cross-condition comparison, (iii) behavioral grounding, (iv) longitudinal analysis, and (v) comparison across saliency methods.

We demonstrate the utility of this methodological framework through three case studies: (i) a comparative analysis of attention trajectories across four classic DRL algorithms (A2C, PPO, DQN, and QR-DQN) in Atari 2600 games, revealing algorithm-specific attention profiles indicative of robustness; (ii) a custom Pong environment designed to isolate the influence of reward shaping, showing how rewards shape both attention and strategy; and (iii) a biomechanical simulation with multimodal sensory inputs including vision and proprioception, where we analyze how attention is distributed across input channels in sequential tasks and diagnose overfitting to redundant sensory channels. Robustness checks across multiple saliency methods further assess the consistency of our findings.

In summary, our contributions include:

- A measurement tool, the hierarchical-attention profile (or h-profile), which quantifies how agents allocate attention to input features by aggregating saliency maps into consistent, structured, and interpretable representations.

- A methodological framework that combines the hierarchical attention profile with longitudinal analysis, controlled comparisons, behavioral grounding, and cross-method replication to support systematic and more reliable analysis of attention allocation during training.

- An empirical analysis of how these attention profiles evolve during training and how they are influenced by key factors: the learning algorithm, the reward function, and the properties of the environment, on three case studies , from Atari 2600 games to biomechanical simulations with multimodal sensory inputs.

- An open-source implementation of our methodology to aid transparency and reproducibility, freely available at `https://anonymous.4open.science/r/attention_profile-F7E6/`

Taken together, these results show that our methodological framework provides a systematic, reproducible and more reliable approach for analyzing how feature reliance emerges, evolves, and is shaped by key factors. By offering a systematic approach to saliency maps we enable the extraction of global attention patterns across algorithms, environments, and reward functions, advancing interpretability beyond isolated case studies of individual agents. In addition to making those systems more transparent, our approach acts as a promising diagnostic axis for identifying algorithmic biases, misallocated attention, and overfitting behaviors invisible to performance metrics alone.

## 2 Related Work

In this section, we review existing work on saliency maps and their application to reinforcement learning, first outlining general methods and then discussing prior efforts to explain RL agents, highlighting the lack of systematic and quantitative analyses that motivates our work.

**Saliency maps** Saliency maps are a class of explainable AI methods designed to highlight the input features that most strongly influence a model's predictions, originally developed to explain predictions of image classifiers in computer vision. They can reveal undesirable effects, such as the Clever Hans effect, where a classifier bases its decision on spurious cues, e.g., a label printed at the bottom of an image rather

than the object itself (Lapuschkin et al., 2019). They can also uncover harmful biases where networks make unfair associations between concepts (Dreyer et al., 2025). Saliency map methods can be broadly grouped into three categories: gradient-based Simonyan et al. (2013); Smilkov et al. (2017) methods such as Grad and SmoothGrad, perturbation-based Greydanus et al. (2018); Puri et al. (2019) methods, and Layer-wise Relevance Propagation (LRP) Bach et al. (2015); Lapuschkin et al. (2019). Gradient-based methods estimate the significance of each input pixel on the network's decision-making process by analyzing the gradient of the output with respect to these pixels. In contrast, perturbation-based methods estimate this influence by altering the input pixels and observing the resulting effect on the network's output. Finally, LRP is a backpropagation-based relevance method that iteratively redistributes prediction scores backward through the network using conservation rules, ensuring that total relevance is preserved. This yields saliency maps that are typically sharper and more localized than those from gradient-based methods, while also being computationally more efficient than perturbation methods, which require multiple forward passes per input (Xing et al., 2023). For these reasons, we adopt LRP as our primary saliency method. To ensure robustness, we also replicate key results using Grad, SmoothGrad, and perturbation methods.

**Explaining RL agents with saliency maps**  Saliency maps have also been applied to deep reinforcement learning agents primarily as an exploratory tool (Zeiler & Fergus, 2014; Zhou et al., 2016; Selvaraju et al., 2017; Weitkamp et al., 2018; Huber et al., 2021; 2022). Most studies in this area focus either on developing new saliency for DRL agents Greydanus et al. (2018); Iyer et al. (2018); Puri et al. (2019); Lapuschkin et al. (2019); Huber et al. (2021),or on designing new agents with built-in saliency using attention units (Nikulin et al., 2019; Annasamy & Sycara, 2019; Mott et al., 2019). In other works, saliency maps are used to interpret potential strategies learned by an agent - either to showcase a new proposed model Markovikj (2022); Wang et al. (2016); Yang et al. (2018), to interpret existing ones Weitkamp et al. (2018) or to illustrate a phenomenon such as observational overfitting where a trained agent relied on spurious but reward-correlated features (Song et al., 2019). Across these studies, saliency maps are typically generated for a single trained agent (or half-trained) on a few state–action pairs, followed by qualitative visual inspection and human interpretation of the agent's strategy. Interpretations that are rarely subjected to experimental validation (Atrey et al., 2019). To the best of our knowledge, only two studies have moved toward a quantitative and systematic analysis of saliency maps to extract broader insights into attention dynamics. Lapuschkin et al. (2019) quantified the evolution of DQN agents' attention during training (via LRP relevance maps) toward objects in Breakout, linking strategy emergence to network depth and replay memory size. Guo et al. (2021) analyzed perturbation-based saliency maps using correlation coefficients and Kullback–Leibler divergences, showing that PPO agents' attention in Atari 2600 games gradually converged toward human gaze patterns, a trend correlated with improved performance. Building on this line of work, our methodology extends these efforts in three key directions: (1) conducting a systematic analysis of attention trajectories across three main factors : reward functions, learning algorithms, and environment dynamics; (2) incorporating behavioral experiments to link attention dynamics to observed agent behavior; and (3) extending the framework to biomechanical visuomotor control tasks, moving beyond image-based analysis toward richer, more realistic scenarios. This advances saliency maps from a primarily exploratory tool to a more systematic instrument for studying attention patterns in DRL agents.

## 3 Methodology

### 3.1 Methodological Framework

Saliency maps have been applied to DRL agents to identify the input features that influenced their decisions. However, their standard representation, a heatmap superimposed on the agent's observation, is high-dimensional, often noisy, and largely dependent on qualitative interpretation. This limits reproducibility, increases the risk of cognitive bias and post hoc interpretation, and makes systematic, longitudinal, and behaviorally grounded analyses difficult.

Nonetheless, saliency maps contain relevant information that remains under-exploited. Here, we propose a methodological framework designed to extract more reliable insights into the emergence and dynamics of DRL agents' attention from saliency maps. The framework relies on five complementary components: (i)

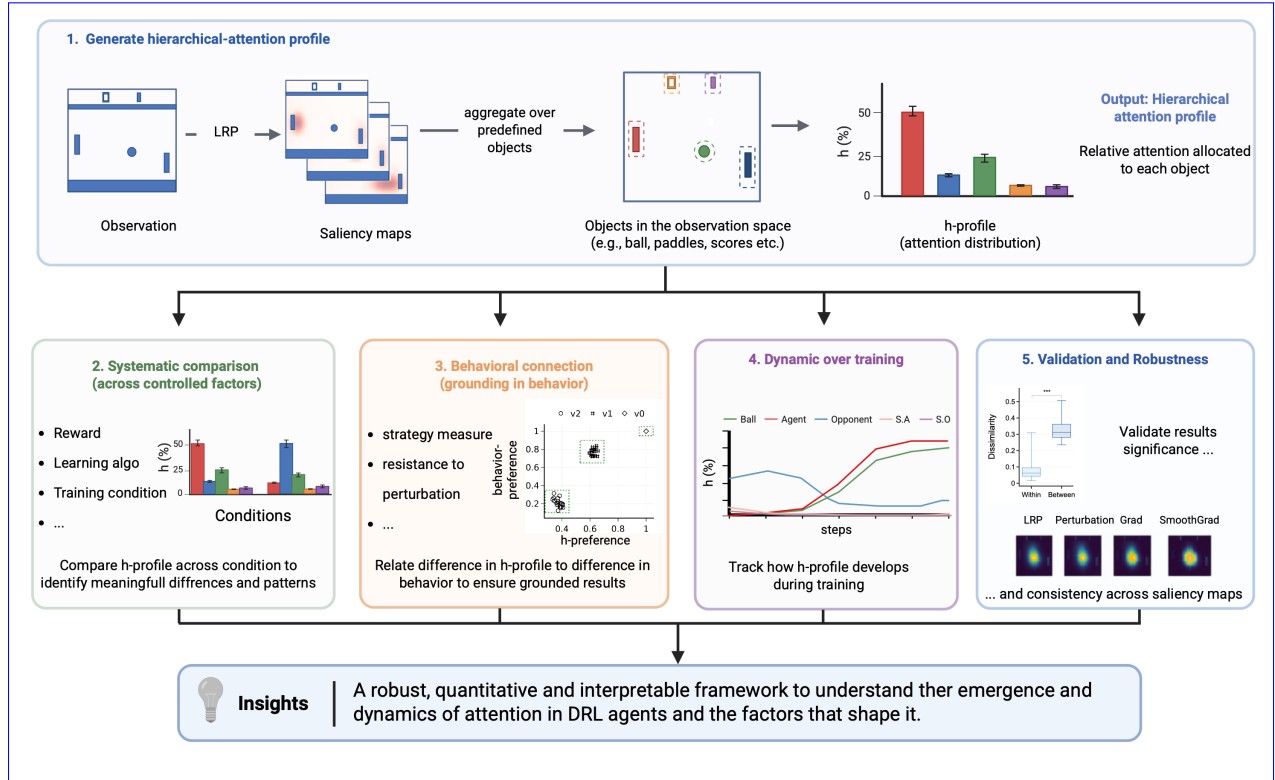

Figure 1: **Illustration of the methodological framework.** Saliency maps are computed over a dataset of inputs before being aggregated into a hierarchical attention profile. This profile is then compared across controlled conditions, connected to behavioral measurements, tracked throughout training, and reproduced with different saliency methods to assess the robustness of the results.

quantification of saliency maps, (ii) cross-condition comparison, (iii) behavioral grounding, (iv) longitudinal analysis, and (v) comparison across saliency methods. An illustration of the methodological framework is shown in Figure 1.

**C1. Quantification** The framework first quantifies saliency maps using a hierarchical attention profile, or $h$-profile, obtained by aggregating pixel-level saliency values over predefined, semantically meaningful entities. This reduces dimensionality, facilitates interpretation, and allows systematic analysis and hypothesis testing. This concept is similar to the approach used by Lapuschkin et al. (2019) to analyze the attention allocated to the bricks object in the Breakout game. Here, this principle is generalized and embedded within a broader methodological framework designed to address several limitations and criticisms associated with the use of saliency maps: their sensitivity to input properties (such as size,color, edge structure), the need for intervention to test saliency-based interpretation Atrey et al. (2019), their limited ability to reflect the learned parameters of the model Adebayo et al. (2018) and, the plurality of saliency methods which can produce diverging heatmaps .

**C2. Cross-condition comparison** The controlled cross-condition comparisons help isolate the factors that influence attention allocation while limiting confounding factors linked to the saliency method itself, such as sensitivity to object size, color, or edge structure.

**C3. Behavioral grounding** Attention profiles alone cannot establish whether attended features are functionally used by the policy. They are therefore combined with behavioral evaluations to determine whether saliency patterns are associated with measurable changes in strategy, robustness, or task performance.

**C4. Longitudinal analysis** Quantifying saliency maps enables tracking their evolution during training. This provides insight into intermediate learning phases, potential learning obstacles, and changes in attention allocation that would be missed by a static analysis of the final model. Furthermore, because the evaluation dataset used to compute the attention profile is constant throughout training, variation in the attention profile can be attributed to changes in the model's weights. This helps distinguish learning-related changes from static artifacts of the input, such as the "edge detector effect" sometimes attributed to saliency methods.

**C5. Comparison with other saliency maps** The framework is independent of any specific saliency extraction method. Since saliency maps lack ground truths and different methods may produce divergent results, cross-methods comparison allows to assess the robustness of the observed attention patterns.

Together, these five components provide a straightforward and interpretable framework for extracting quantitative summaries of saliency maps, analyzing attention dynamics, and assessing behavioral feature reliance while mitigating the risks commonly associated with saliency-based explanations.

In the next section, the computation of the hierarchical-attention profile from LRP is presented. The extension to gradient-based, and perturbation-based methods is presented in the Appendix B.

### 3.2 Hierarchical-attention profile

Saliency maps identify input features influencing an agent's prediction, but their raw format hinders their systematic analysis. We propose the hierarchical-attention profile ($h$-profile) to quantify them at the object level. Given a predefined set of objects in the input space, the $h$-profile aggregates feature-level saliency within each object and expresses its importance relative to others. For instance, in Atari 2600 Pong, the saliency of all pixels forming the paddle can be grouped to measure its overall relevance. In MuJoCo tasks Todorov et al. (2012), the same principle applies to continuous variables such as joint angles. By mapping pixel- or feature-level saliency to interpretable objects, this profile offers a straightforward yet effective means to obtain consistent, interpretable, and quantitative summaries of saliency maps, facilitating the systematic analysis of agent attention.

Since the $h$-profile is a post-hoc quantification of saliency maps, it can be derived from any saliency method, such as gradient- or perturbation-based approaches. We primarily used Layer-wise Relevance Propagation, as it produces sharp saliency maps with strong focus on image objects (Appendix L.1). For a model output $f(\boldsymbol{x})$, LRP calculates a relevance score for each neuron in the previous layer. Due to its conservation property, relevance can be propagated layer-by-layer up until the input, resulting in a relevance map per layer. To ensure robustness, we also reproduced our key results using Grad, SmoothGrad, and perturbation methods (Simonyan et al., 2013; Smilkov et al., 2017; Greydanus et al., 2018). Detailed derivations of the $h$-profile for each saliency method are provided in Appendix B, and the corresponding reproduction of results is reported in Appendix L.

The $h$-profile is evaluated on a dataset $\boldsymbol{X}$ consisting of environment inputs paired with object-level labels $\bar{\boldsymbol{X}}$. These labels specify the mapping between input features and the objects defined for the study—for example, pixels assigned to the ball or paddle in Pong, or inputs assigned to the vision versus proprioception channels in the visuomotor biomechanical tasks. Each sample must satisfy two conditions: (i) all objects selected for analysis are present, and (ii) in image inputs, objects do not overlap. Details of dataset construction and object labeling for each environment are provided in Appendix C. The dataset may be regenerated at each evaluation or fixed throughout training. To ensure consistent comparison of $h$-profiles across settings, algorithms, and training steps, we use a fixed dataset. In Appendix C, we empirically compare both approaches, which show similar results.

Relevance maps for the $h$-profile were generated from the final hidden layer noted $F_c$, which encodes the representation preceding the output. As neurons in this layer capture different concepts and vary in relevance (cf. concept relevance propagation Achtibat et al. (2023); Dreyer et al. (2025)), we computed neuron relevance scores and retained the subset accounting for $p = 90\%$ of total relevance. Relevance maps were then generated for these neurons at the model input.

For each input $\boldsymbol{x} \in \boldsymbol{X}$, we compute the $F_c$ neurons relevance score with respect to the output $\{R_i^{(\text{output})}(\boldsymbol{x})\}_{i \in F_c}$ and select a subset of neurons $S \subseteq F_c$ that account for $q \geq 90\%$ of total relevance $R_{F_c}^{(\text{output})}(\boldsymbol{x}) = \sum_{i \in F_c} R_i^{(\text{output})}(\boldsymbol{x})$. For each neuron $k \in S$, input-level relevance maps are given by $\boldsymbol{R}^{(k)}(\boldsymbol{x}) = \{R_p^{(k)}(\boldsymbol{x})\}_{p \in \boldsymbol{x}}$ with $p$ a feature of the input $\boldsymbol{x}$. Details regarding the derivation of the relevance scores and the propagation rules used for each case study are provided in Appendix A.

Given a predefined finite set of objects, $O$, we define the *hierarchical-attention* profile, $h : O \to \mathbb{R}$, as:

$$h(o_j) = \frac{1}{|\boldsymbol{X}|} \sum_{\boldsymbol{x} \in \boldsymbol{X}} \sum_{k \in S} R_k^{(\text{output})}(\boldsymbol{x}) \cdot R_j^{(k)}(\boldsymbol{x}), \tag{1}$$

$$R_j^{(k)}(\boldsymbol{x}) = \sum_{p \in P_{o_j}} R_p^{(k)}(\boldsymbol{x}) \tag{2}$$

where $P_{o_j}$ is the set of input features belonging to object $o_j \in O$ according to the mapping defined by $\bar{\boldsymbol{X}}$, e.g., set of pixels belonging to object $o_j \in O$. For vector inputs, objects correspond to feature indices, and $R_j^{(k)}(\boldsymbol{x})$ reduces to the relevance at index $j$. The $h$-profile thus quantifies the average relevance allocated to each object across inputs and neurons, yielding a measure of the agent's attention distribution. The impact of the size of the dataset $N$ and the threshold on the total relevance $q$ (here $N = 50$ and $q = 90\%$) are discussed in Appendix D. Fig. 5 show how the $h$-profile is computed using the LRP method.

### 3.3 Experimental Setup

To illustrate the breadth of our methodology, we analyzed agent attention in three case studies of increasing complexity, ranging from benchmark environments to real-world DRL applications: two pixel-based environments (Atari 2600 and Custom Pong), and a muscle-driven biomechanical environment with multimodal inputs including proprioception and vision. For each case study, we describe the environment, define the objects used for computing the $h$-profiles, and detail the protocols used to assess the agent's behavior. Annotated objects are listed in Table 1, with full implementation details for the games in Appendix H.

#### 3.3.1 Case Study I: Atari 2600

Our first case study examines attention patterns across algorithms (A2C,PPO,DQN,QR-DQN) on the standard Atari 2600 benchmark. We analyze how different learning algorithms shape agent attention, and illustrate how these patterns relate to the algorithm's robustness under environmental perturbations using Breakout as an example.

**Environment and Objects** We evaluated four Atari 2600 games from the Gymnasium library (Brockman et al., 2016; Towers et al., 2024): Breakout, Pong, Space Invaders, and Ms. Pacman. For each game, the annotated task-relevant objects used for $h$-profiles can be found Table 1 and an illustration of the objects for each game can be found in Appendix C Figure 7.

**Behavioral Measurement: Perturbation Test** To assess the link between the difference in attention patterns between learning algorithms and their robustness to perturbation of their environment, we introduced visual perturbations in Breakout (color changes to bricks, and score display plus wall above to have a similar number of pixels perturbed). Performance difference was measured as : $\Delta r = \frac{r_{perturbed} - r_{unaltered}}{r_{unaltered}}$, with $r$ the average reward obtained over 10 evaluation episodes on the unaltered and the perturbed environments. Here, $\Delta r \sim -1$ indicates near-complete failure in the perturbed environment (minimum reward = 0), $\Delta r \sim 0$ indicates comparable performance across environments, and $\Delta r \sim 1$ indicates improved performance under perturbation. Robustness was evaluated on all 10 agents of each algorithm every 1M training steps.

#### 3.3.2 Case Study II: Custom Pong

The second case study investigates how induced strategies through different reward design can shape an agent's attention and how these differences manifest in agent behavior.

**Environment and Objects**  We implemented three Pong variants with identical visuals but distinct reward functions to measure the impact of the reward functions on the agent's attention profile:

- **Baseline Pong (v0)**: standard gameplay, rewards for scoring/missing the ball

- **Distractor Pong (v1)**: introduces a second ball (B2) with no reward signal

- **Dual Pong (v2)**: both balls yield rewards: B1 as in v0, B2 rewards paddle rebounds

**Behavioral Measurement: Dual Ball Discrimination Test**  To evaluate the connection between attention patterns for each ball and the preference of the agent for each ball we put 20 agents to a Dual Ball Discrimination Test. Trained agents were evaluated on 100 controlled trials in which they could interact with either *B1* or *B2*, but not both simultaneously Preference was quantified as choice frequency; a color-swap of B1 and B2 control tested for pixel value reliance. For more details about the experiment see Appendix J.1.

### 3.3.3  Case Study III: Visuomotor Biomechanical Tasks

The third case study explores how muscle-driven agents allocate attention across sensory modalities (proprioception, vision) in typical Human-Computer interaction tasks from the User-in-the-Box repository Ikkala et al. (2022), providing a more realistic setting for the application of our methodology.

**Environment and Objects**  We used the provided *MoBL Arms Model* Saul et al. (2015), a muscle-actuated biomechanical model implemented in OpenSim Delp et al. (2007), comprising 26 muscles and 5 joints. This model interacts with a simulated environment via the MuJoCo physics engine Todorov et al. (2012), receiving perceptual feedback through multiple sensory channels including: vision, proprioception, and game-state inputs. Agents were trained on two tasks:

- **Choice reaction task**: the environment consists of a display and four colored buttons. At the start of each trial, the display color indicates which button to press. A trial ends either when the agent presses the correct button with the required force or after a 4-second timeout. One episode consists of 10 trials.

- **Parking a remote control car**: the agent controls a joystick on a gamepad to park a car by braking/accelerating. One episode consists of one trial, and either ends after 10 seconds or when the car is parked in the marked area.

Annotated objects for computing the *h*-profile depend on the analysis level. When comparing attention across sensory modalities, the objects corresponded to the vision and proprioception channels. A finer-grained analysis, separating each element of the vision channel and each proprioceptive feature, is provided in Appendix K.1.

**Behavioral Measurement**  Performance was measured as the average proportion of correct button presses in the choice reaction task, and as the proportion of episodes in which the car is inside the target (successful parking) for the parking a remote control car task, each evaluated over 10 episodes. To probe overfitting in the choice reaction task, we implemented a Moving Buttons variant where button positions shifted $10cm$ randomly during training. Attention toward the buttons was then compared between agents trained on the static vs. the moving version. In this study, each proprioceptive input dimension was treated as a separate object, and the visual stream was decomposed into buttons, screen, and the biomechanical arm.

### 3.4  Agents

DRL algorithms were taken from the Stable Baseline 3 repository Raffin et al. (2021). For the Atari 2600 case study we trained A2C Mnih (2016); Konda & Tsitsiklis (1999), PPO Schulman et al. (2017), DQN Watkins & Dayan (1992); Mnih et al. (2013) and QR-DQN Dabney et al. (2018) agents. All agents used identical network architectures. Algorithm identity was defined as the combination of learning rule and

Table 1: Annotated objects used for computing $h$-profiles across environments.

| Case Study | Environment | Objects |
|---|---|---|
| Atari 2600 | Breakout | Ball, agent paddle, bricks, agent score (S.A.) |
| | Pong | Ball, agent paddle, opponent paddle, agent score (S.A.), opponent score (S.O.) |
| | Space Invaders | Agent, hiding spots (H.S.), shots, aliens, agent score (S.A.) |
| | Ms. Pacman | Agent, ghosts, lives, fruits, agent score (S.A.) |
| Custom Pong | | Ball (B1), ball (B2), agent paddle, opponent paddle, agent score (S.A.), opponent score (S.O.) |
| Biomechanical | Choice Reaction | Vision channel (buttons, screen, arm), proprioception channel (joint angles, forces) |
| | Car Parking | Vision channel, proprioception channel |

associated hyperparameters, following standard benchmarking practice. For the Custom Pong study case we used A2C agents. For the biomechanical task we used PPO agents using a multi-input policy architecture. The policy network integrated observations via two encoders and a shared actor–critic module. Details about all network architectures and hyperparameters can be found in Appendix E and F, respectively.

The $h$-profile is computed from the same penultimate layer, denoted $Fc$, for all four algorithms: A2C, PPO, DQN, and QR-DQN. For A2C and PPO, the actor and critic share the same network up to $Fc$, after which separate heads predict the policy and the value. Thus, the $F_c$ representation used to compute the $h$-profile is shared by both the actor and the critic. We chose this layer because it provides a comparable representational bottleneck across algorithms: although it is shaped by different learning objectives (policy/value losses for A2C and PPO, Q-value prediction for DQN, and distributional return prediction for QR-DQN), it sits immediately upstream of action selection and is therefore directly relevant to agent behavior.

### 3.5 Statistical Tests

To assess group-level differences in attention profiles (Atari 2600 and Custom Pong), we used Analysis of Similarities (ANOSIM), a non-parametric rank-based test implemented in scikit-bio (Rideout et al., 2025). Significance was estimated via 99 permutations, and the $R$ statistic reported the degree of separation ($0 =$ no separation, $1 =$ complete separation).

For the biomechanical setting, differences in attention toward the buttons between agents trained in static vs. moving environments were tested with a two-sample t-test. Normality and equality of variances were verified using Shapiro–Wilk and Levene's tests, respectively. Statistical analyses were conducted with JASP (JASP Team, 2025).

## 4 Results

We apply the proposed $h$-profile to trace how agents allocate attention over the course of learning. Results are presented through three case studies of increasing complexity: Atari 2600 benchmarks, Custom Pong variants, and visuomotor interactive tasks. Each case isolates a distinct factor shaping attention and its link to behavior: the role of the learning algorithm and robustness to perturbations, the impact of reward design and emergence of unintended strategies, and the dynamics of multimodal inputs with potential overfitting to a single channel. Across these settings, we systematically examine (i) how attention evolves during training, (ii) whether stable, algorithm- or task-specific patterns emerge, and (iii) how these patterns translate into behavioral outcomes.

### 4.1 Case Study I:Atari 2600 – Attention dynamics reveal algorithm-specific representational biases and vulnerabilities to perturbations

In our first case study, we examine whether different DRL algorithms develop distinct attention trajectories under the same tasks, and whether these reflect their robustness to environmental changes (Section 3.3.1).

**Attention development during training**  Fig. 2 **a-b** illustrates the reward and attention trajectories in Breakout. Across all algorithms, early training showed similar attention patterns dominated by the task structure: agents quickly reduce their attention to the bricks and the score while increasing their attention to the ball and the agent's paddle. This redistribution of agent's attention occurred before noticeable performance gains. The same early dynamics were observed in Pong, Space Invaders, and Ms.Pacman (Appendix I.2). Over time, however, algorithm-specific trajectories emerged, even if performance (score) was similar. DQN and QR-DQN progressively redirected attention back to the bricks ($h \sim 60\%$), whereas A2C and PPO agents maintained lower focus ($h \sim 20\%$). Notably, DQN was unique in showing an early increase in attention to the bricks, coinciding with a performance plateau at lower final scores ($\sim 200$) compared to A2C, PPO, and QR-DQN ($\sim 400$). This divergence is partly explained by the use of replay buffers in value-based methods, which bias attention toward brick-related features (see Appendix I.3).

**Linking attention to behavior: robustness under perturbation**  In Breakout, altering the color of the bricks ,a feature heavily attended by DQN and QR-DQN , caused severe degradation, with $\Delta r$ approaching ˘1 during training. In contrast, A2C and PPO, which allocated little attention to bricks, remained robust ($\Delta r \approx 0$). Control perturbations affecting non-attended features (score display, wall color) had no measurable effect on any algorithm (Fig. 2 **c**).

**Stable algorithm-specific differences**  The differences in the $h$-profile were not limited to Breakout. Across all four games, attention profiles diverged systematically between algorithms even at matched performance levels (Fig. 2 **d**–**e**). To test this formally, we computed pairwise Euclidean distances between $h$-profiles and applied Analysis of Similarities (ANOSIM) with algorithm as the grouping factor. Significant differences were found in every game ($p = 0.01$ for all games, $R_{pong} = 0.80$, $R_{breakout} = 0.90$, $R_{spaceinvaders} = 0.88$, $R_{pcman} = 0.93$, sample size = 40, 99 permutations). Details of the pairwise dissimilarity between learning algorithms can be found Appendix I.1.

The early similarity of attention dynamics across algorithms, the significant dissimilarity in final $h$-profiles, and the over-reliance of DQN and QR-DQN on bricks were all replicated with the three alternative saliency methods (Appendix L.2).

**Together, these results show that our attention profile can expose representational biases arising from both algorithmic design and hyperparameter choices (e.g., replay buffer size), and reveal vulnerabilities to perturbations invisible to performance evaluations alone.**

### 4.2 Case Study II: Custom Pong - Attention dynamics uncover (unintended) strategies arising from reward design

Our second case study intends to isolate effects of induced strategies through reward design on the agent's attention. As Atari 2600 environments provide limited experimental control, which makes it difficult to isolate the specific task factors that shape an agent's attention, we designed three custom Pong variants with similar visual inputs but distinct reward structures: Baseline Pong, Distractor Pong, and Dual Pong (Section 3.3.2).

**Attention development during training**  We tracked $h$-profiles for ten A2C agents per Pong variant throughout training. Across all versions, attention followed a consistent developmental sequence: initial focus on the displayed scores, then on the moving ball(s), followed by the opponent's paddle, and finally the agent's own paddle (Fig. 3 **a-b**). Notably, attention to the agent's paddle emerged late and coincided with performance improvements, suggesting that attention shifts aligned with learning progress. Early on,

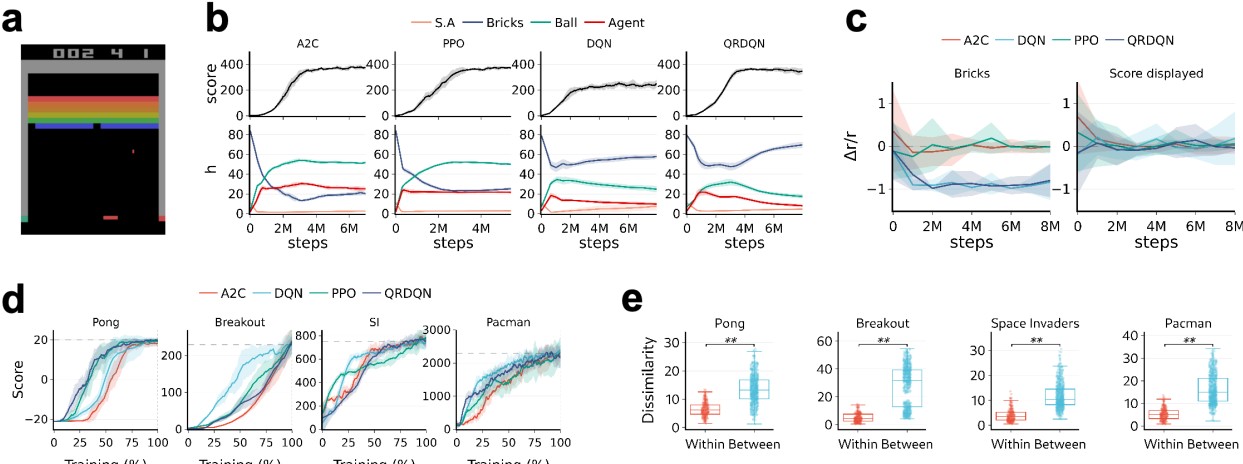

Figure 2: **Attention dynamics reveal algorithm-specific differences in Atari 2600 games.** **a.** Breakout game. **b.** Breakout training curves show performance (top) and hierarchical attention profiles (bottom), tracking the proportion of attention allocated to game objects (ball, paddle, bricks, score agent (S.A)). Early convergence toward the ball is followed by algorithm-specific divergence, with DQN and QR-DQN reallocating substantial attention to bricks. **c.** Robustness under perturbations: modifying brick color severely degrades DQN/QR-DQN performance ($\Delta r \to -1$), while A2C and PPO remain robust; altering irrelevant features (score display, wall) has little effect on all algorithms. **d.** Learning curves across four Atari games and four learning algorithms. Standard deviation is represented by a shaded area. **e.** Dissimilarity analysis shows attention profiles are significantly more similar within algorithms than between algorithms (ANOSIM, ** indicates $p < 0.01$).

visually similar objects (balls, paddles) were not clearly differentiated, but distinctions sharpened as training advanced.

**Reward-dependent final allocations**  Final attention distributions reflected the reward structure of each environment. Agents in Single-Ball Pong and Distractor-Ball Pong concentrated on B1, the rewarding ball, while agents in Dual-Ball Pong shifted their focus to the distracting ball B2. Importantly, in the Distractor-Ball setting, agents continued to pay attention to B2 despite its lack of reward association.

To quantify these differences, we compared final $h$-profiles across 50 agents per variant (Fig. 3 **c**) using ANOSIM. Since B2 was absent in Single-Ball Pong, the analysis focused on Distractor-Ball vs. Dual-Ball agents. Profiles were significantly more similar within conditions than across them ($R = 0.97$, $p = 0.01$, sample size $= 100$, 99 permutations; Fig. 3 **d**), confirming that the reward structures shaped attention.

**Linking attention to behavior: interaction preferences**  To test whether attentional differences translated into behavior, we submitted 20 trained agents per variant to the Dual Ball Discrimination Test (Section 3.3.2). In 100 controlled trials, agents consistently interacted most with the ball that had received the most attention (Fig. 3 **e**). Color-swapping did not alter these preferences (Appendix J.1), confirming that choices reflected learned relevance rather than low-level cues. Crucially, Distractor-Ball agents still engaged with B2, mirroring their residual attention to it.

The consistency of $h$-profiles within each game variant, as well as the alignment between attentional focus on a ball and its associated reward structure, were replicated across the three alternative saliency methods and can be found in Appendix L.3)

**These results show that reward functions shape agents' *attention*, and that differences in attention translate into behavioral differences. Moreover, attention profile can diagnose when agents misallocate attention to irrelevant cues and develop unintended strategies arising from the reward design.**

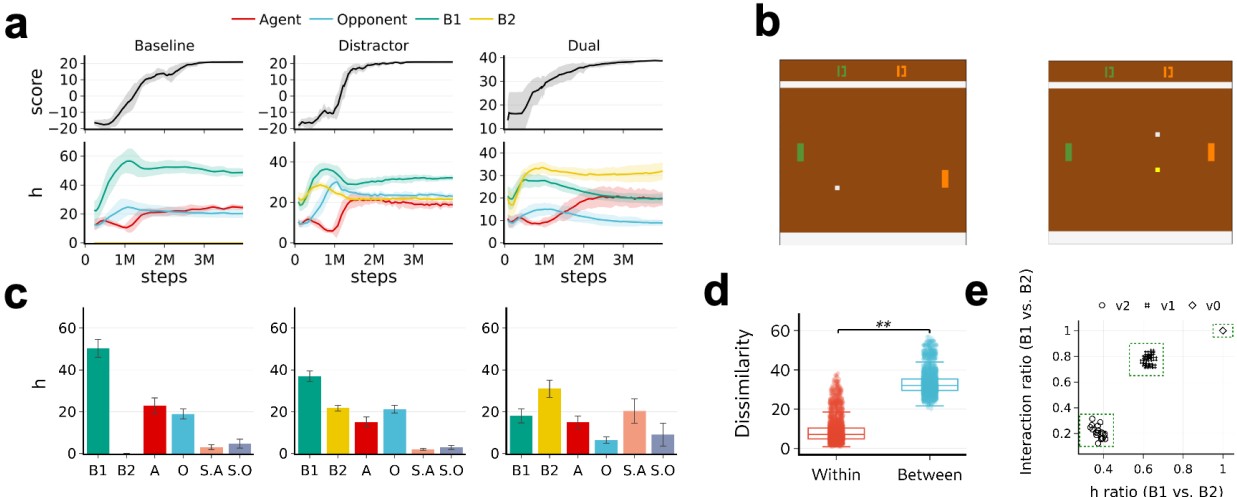

Figure 3: **Reward structure shape distinct attention allocation in Custom Pong. a.** Training curves show performance (top) and hierarchical attention profiles (bottom), tracking the proportion of attention allocated to game objects (Ball 1, Ball 2, Agent, Opponent). Agents in the Distractor (v1) version focused primarily on the rewarding ball (B1), whereas agents in the Dual-ball condition (v2) shifted attention to B2. Notably, v1 agents continued to allocate attention to the distractor ball (B2). **b.** Pong environments with one ball (left) and two balls (right). The white and yellow balls denote B1 and B2, respectively. **c.** Hierarchical-profile $h$ at the end of training for the balls (B), the agent's paddle (A), the opponent's paddle (O), the score of the agent displayed (S.A), and the score of the opponent displayed (S.O). The bar represents the average over 50 agents, the standard deviation is represented as an error bar. **d.** Dissimilarity analysis on the h-profile of trained agents from v1 and v2 shows that attention profiles are significantly more similar within game version than between (ANOSIM, ** indicates $p < 0.01$). **e.** In a dual-ball discrimination test, the ball receiving more attention was also the one most frequently interacted with (v0 was added as a reference).

### 4.3 Case Study III: Visuomotor Interactive Tasks - **Attention shifts across vision and proprioception and can help detecting overfitting to redundant cues**

While Atari 2600 and custom Pong environments allow us to probe algorithmic and reward-driven influences on attention, we want to extend our research from 2D pixel space to more real-world like scenarios. Real-world applications (robotics, assistive devices, human interaction) require agents to integrate multiple sensory modalities and operate in sequential tasks structure. To capture these demands, we extended our analysis to PPO agents trained in biomechanical control environments. This setting allows us to examine how attention is reallocated both across modality and during different phases of sequential task completion (Section 3.3.3).

**Dynamic allocation of attention during sequential learning** For each task, we trained three agents with a different seed. We measured their attention profile, their task performance (pressing the right button or parking the car) and their performance score during training (Section 3.3.3). In the parking a remote control car task (Fig. 4 **c**), attention shifted systematically from proprioception to the visual channel. During the initial 'reaching phase', when rewards were primarily based on the progress of the arm towards the joystick, there was an increase in attention towards proprioceptive inputs. This was then followed by a "completion phase" where visual inputs (car–target alignment) became increasingly prioritized as rewards became tied to successful task completion (parking the car in the target area). In the choice reaction task (Fig. 4 **d**), early training demonstrated consistent attention to both proprioceptive and visual inputs. However, over time, vision gradually became more important, coinciding with higher rewards for correctly pressing the target buttons.

**Linking attention to behavior: moving buttons** In the choice reaction task, agents systematically allocated more attention to proprioceptive input than to visual input, whereas the opposite pattern was

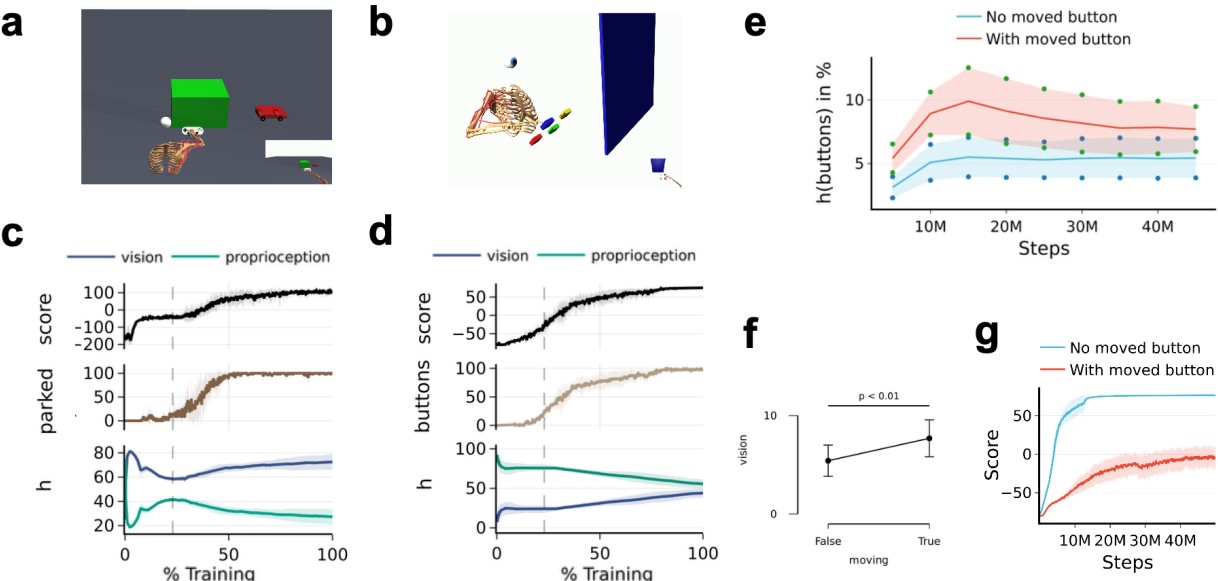

Figure 4: **Dynamic reallocation of attention across modalities. a.** Parking a remote control car task. **b.** Choice reaction task. **c. and d.** Performance (top), task success (middle: % parked or % correct button presses) (middle) and hierarchical attention profiles (bottom) during training for the parking a remote control car task **c** and the choice reaction task **d**. In both tasks, attention to vision increases when reward increased due to visually guided task completion (dotted line). Lines show mean ± SD over 10 agents. **e.** Agents trained on the choice reaction task with moving buttons pay more attention to the displayed buttons than those trained with static buttons. **f.** A t-test confirmed the significant difference between those two groups. **g.** Training performance was higher for agents with static buttons, reflecting the greater difficulty of the moving-button condition.

observed in the parking a remote control car task. This imbalance could be indicative of information redundancy: with static button positions, focusing on proprioception and looking at the screen (but not the buttons) can suffice to guide actions, similar to using muscle memory. In order to ascertain whether agents overfit to a redundant input channel, we trained agents in a modified environment where button positions shifted randomly during training (Section 3.3.3). We found that these agents allocated significantly more attention to the visual input (Fig. 4 **e-f**), as confirmed by a [paired](#) $t$-test ($p = 0.009$, Cohen's $d = -1.31$) on the agents' attention on the buttons after 50M steps. (Assumption checks indicated that both groups were normally distributed, Shapiro–Wilk: static $W = 0.97$, $p = 0.87$; moving $W = 0.95$, $p = 0.56$, and had equal variances Levene's $F(1, 18) = 0.42$, $p = 0.52$ ). However, agents trained in the modified environment had lower performance scores ($-4.5 \pm 12.0$) than those trained in the original environment ($76.4 \pm 0.25$), reflecting the increased difficulty of the moving task (Fig. 4 **g**). This supports the interpretation that the earlier reliance on proprioception could be an instance of overfitting to redundant cues. The increase in attention on the buttons in the moving task compared to the static task was also found with the gradient method, see [Appendix L.4](#).

**These results show that agents dynamically reallocate their attention towards the modality most impactful for reward at different stages of learning. Moreover, attention profiles can capture adaptive cross-modality shifts in information use and help detect overfitting to sensory cues when information in an environment is redundant.**

## 5 Discussion

Across three case studies of increasing complexity, we demonstrate that attention profiles can serve as diagnostic tools for understanding agent behavior and learning dynamics. Our saliency-derived *h-profiles* reveal representational biases, vulnerabilities, and strategies that are not visible from performance scores alone.

Even when algorithms achieved comparable rewards, their attention profiles diverged, exposing algorithm-specific vulnerabilities. Perturbation experiments confirmed these differences, as performance dropped when attended features were disrupted. This suggests that benchmark evaluation should move beyond reward alone to include the diversity of attention patterns as an additional axis of comparison for DRL algorithms.

Reward structures also strongly influenced what agents prioritized. In the Distractor-Ball setting, for example, agents allocated attention to irrelevant cues, and behavioral testing showed that these attentional biases shaped action choices. Attention profiles therefore provide an early safeguard for detecting unintended strategies induced by reward design—patterns that performance metrics alone may overlook.

In visuomotor interactive tasks, attention analysis revealed both adaptive strategies and problematic over-reliance. For instance, in the Choice Reaction task, agents relied on proprioception because of redundant cues. Such reliance is not inherently negative but may fail under missing or unreliable channels. Detecting these biases highlights the potential of attention profile for environment design to create more flexible and generalizable agents.

Our methodological framework generalizes across environments and saliency methods, offering a scalable approach for tracking representational development in DRL. However, it also has limitations. In particular, it relies on predefined object or modality labels . While this makes the analysis straightforward in structured inputs with semantically interpretable signals (e.g., proprioceptive channels in the UITB environment), it is less applicable to raw, high-dimensional inputs such as pixel observations, which first require an explicit abstraction step. The choice of abstraction can introduce bias into the representation being analyzed and may obscure relational features, such as distances or velocities , that are more difficult to label explicitly. Nevertheless, these limitations point to clear directions for future work. Combining our methodology with automatic segmentation tools (e.g., SAM Kirillov et al. (2023)), relational abstractions Jansma (2025), or causal probing methods could extend attention analysis beyond object-level attribution.

Our findings extend prior work demonstrating consistency of saliency patterns within algorithms by revealing systematic differences across algorithms, reward functions, and environments (Guo et al., 2021; Lapuschkin et al., 2019). Attention profiles thus emerge as stable signatures of algorithmic behavior, linked to robustness against perturbations and sensitive to reward design. Extending our analysis across saliency methods echoes earlier comparative studies, with the type of saliency methods chosen having some influence over an agent's attention profile (Adebayo et al., 2018; Yona & Greenfeld, 2021; Hedström et al., 2024). However, tracking their evolution during training reveals a consistent dynamic signal, and when applied to constrained, well-defined questions and validated with behavioral experiments, the different saliency methods provided consistent insights into agent behavior.

Looking ahead, attention trajectories could be used to predict robustness and generalization, inform algorithm selection, and guide reward shaping or curriculum design. Extending these tools to robotics and assistive systems could facilitate interpretability in these high-stakes domains.

Finally, parallels with human attention research offer promising avenues. Cognitive neuroscience shows that attention can be shaped by both reward and stimulus saliency—tendencies we also observe in DRL agents (Failing & Theeuwes, 2018; Desimone et al., 1995). Without assuming shared mechanisms, adopting controlled manipulations and longitudinal analyses from neuroscience could deepen our understanding of machine attention.

In sum, attention profiles offer more than post hoc explanations: they provide a window into the learning process itself. By revealing what agents attend to, and how this evolves during training, we can design DRL agents that are not only high-performing but also robust, interpretable, and aligned with human expectations.

## 6 Conclusion

We demonstrated that saliency-derived attention profiles uncover aspects of learning in DRL agents that remain hidden behind performance scores, exposing algorithmic biases, unintended consequences of reward design, and overfitting. By aggregating saliency into object- and modality-level profiles, our hierarchical profile provides a scalable methodology for tracking how agents prioritize information during train-

ing. Embedded within a broader methodological framework combining longitudinal analysis, controlled comparisons, behavioral grounding, and cross-method saliency replication, this profile supports more systematic and reproducible analysis of saliency maps in DRL agents. These findings position attention analysis as a powerful diagnostic tool: it can reveal vulnerabilities, inform task and reward design, and guide the development of more robust and interpretable agents. Looking forward, our methodology suggests a path toward a principled "science of machine attention." By combining controlled experiments, robust attention profiles, and insights from cognitive neuroscience, future research could not only track what agents learn, but actively shape it. Attention analysis thus offers a promising diagnostic axis, alongside performance and generalization, for advancing the development of DRL agents that are interpretable, robust, and reliable in real-world settings.

## Impact Statement

This paper presents work whose goal is to improve the transparency and reliability of DRL agents, ultimately promoting more trustworthy use of deep RL.

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

# A  LRP relevance score derivation

Let $f : \mathbb{R}^n \to \mathbb{R}^m$ be the feedforward neural network characterizing an agent's policy, where $\boldsymbol{x} \in \mathbb{R}^n$ is the input image to the network, and $f(\boldsymbol{x}) \in \mathbb{R}^m$ is its output. The function $f$ can be decomposed as

$$f(\boldsymbol{x}) = f_L \circ f_{L-1} \circ \cdots \circ f_1(\boldsymbol{x}),$$

with each $f_{l \in L}$ representing a transformation at layer $l$.

Applied to an input image $\boldsymbol{x}$, LRP Bach et al. (2015); Lapuschkin et al. (2019) calculates a relevance score $R_p(\boldsymbol{x})$ for each pixel $p \in \boldsymbol{x}$, indicating its contribution to the network's decision. The output of LRP is a relevance map (or heatmap), denoted

$$\boldsymbol{R}(\boldsymbol{x}) = \{R_p(\boldsymbol{x})\}_{p \in \boldsymbol{x}}.$$

This is derived by iteratively backpropagating the output $f(\boldsymbol{x})$ from layer $l + 1$ to layer $l$ according to propagation rules defined by a relevance model Montavon et al. (2017). The general formulation is:

$$R_{i \in l} = \sum_{j \in l+1} \frac{q_{ij}}{\sum_{i' \in l} q_{i'j}} R_j, \tag{3}$$

where $R_i$ is the relevance of neuron $i \in l$ and $R_j$ is the relevance of neuron $j \in l + 1$. The propagation coefficients $q_{ij}$ depend on the chosen rule, which is adapted to the input domain (Montavon et al., 2019).

In our experiments: - For the Atari 2600 and Custom Pong tasks, inputs are pixel values or ReLU activations, both non-negative. We therefore used the $\alpha\beta$-rule with $\alpha = 1$ and $\beta = 0$ (Bach et al., 2015). - For the biomechanical model experiment, inputs may be negative and activations come from LeakyReLU. In this case, we used the $\alpha\beta$-rule with $\alpha = 2$ and $\beta = 1$. - In all experiments, for the first layer we used the $\omega^2$-rule (Montavon et al., 2017).

The corresponding propagation rules are:

**The $\alpha\beta$-rule**

$$R_j^{(l)} = \sum_k \left( \alpha \frac{(a_j w_{jk})^+}{\sum_{0,j}(a_j w_{jk})^+} - \beta \frac{(a_j w_{jk})^-}{\sum_{0,j}(a_j w_{jk})^-} \right) R_k^{(l+1)} \tag{4}$$

**The $\omega^2$-rule**

$$R_i^{(l)} = \sum_j \frac{w_{ij}^2}{\sum_i w_{ij}^2} R_j^{(l+1)} \tag{5}$$

## A.1  Stepwise procedure

To evaluate which regions of an input $\boldsymbol{x}$ contribute to the activation of a neuron in the $F_c$ layer, we apply the following steps for each $\boldsymbol{x} \in \boldsymbol{X}$:

**1. Compute neuron relevance scores in $F_c$**   We backpropagate the output relevance $\boldsymbol{R}^{(\text{output})} = f(\boldsymbol{x})$ to the layer $F_c$ using the rules in Eqs. (4) and (5).This yields the set $\{R_i^{(\text{output})}(\boldsymbol{x})\}_{i \in F_c}$. The subset $S \subseteq F_c$ is then defined as the smallest set of neurons covering at least $q\%$ of the total relevance:

$$R_{F_c}^{(\text{output})}(\boldsymbol{x}) = \sum_{i \in F_c} R_i^{(\text{output})}(\boldsymbol{x}).$$

**2. Generate input-level relevance maps.**   For each neuron $k \in S$, we initialize the relevance vector in layer $F_c$ as $\boldsymbol{R}_{F_c} = (t_1, \ldots, t_k, \ldots, t_N)$,   with $t_i = \begin{cases} 1, & \text{if } i = k, \\ 0, & \text{otherwise.} \end{cases}$

This relevance is backpropagated to the input layer using the rules in Eqs. (4) and (5), producing the input-space relevance maps $\boldsymbol{R}^{(k)}(\boldsymbol{x}) = \{R_p^{(k)}(\boldsymbol{x})\}_{p \in \boldsymbol{x}}$. The resulting maps highlight input regions most relevant to

each neuron $k$ in $F_c$. This process yields relevance maps aligned with the labeled input $\bar{\boldsymbol{x}}$, revealing which regions of the input the network attends to. For implementation, we used the `Zennit` library (Anders et al., 2021).

**3. Extension to the biomechanical task.** For the biomechanical models (Section 4.3), the environment setup enabled us to extend LRP to regression tasks, where explanations are meaningful when outputs are defined relative to a reference (Letzgus et al., 2022). In our case, the output represents relative muscle activation: whether a given muscle should remain constant ($y = 0$), increase ($y > 0$), or decrease ($y < 0$), with $-1 \leq y \leq 1$. Accordingly, we use $y_{\text{ref}} = 0$ as the natural reference value.

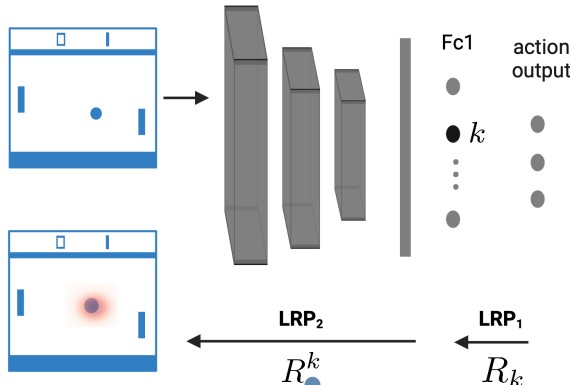

Figure 5: Illustration of extracting the relevance score for the ball object from neuron $k$ in the $F_c$ layer. Only one of the four input frames is shown for clarity. The first LRP pass ($LRP_1$) computes neuron relevance, and the second ($LRP_2$) computes object relevance with respect to neuron $k$.

# B    Validation of our results: comparison with different saliency methods

In addition to LRP, we evaluated several established saliency extraction methods: gradient-based saliency, SmoothGrad, and perturbation-based methods. For gradient and SmoothGrad, we used the implementations provided in Zennit (Anders et al., 2021). For the perturbation-based method, we adapted the code from Greydanus et al. (2018).The final saliency maps were obtained by taking the absolute value of the raw saliency maps and normalizing them.

## B.1    Gradient-based saliency

Gradient-based saliency maps Baehrens et al. (2010); Simonyan et al. (2013) are computed from the gradient of a target output $y$ with respect to the input $\boldsymbol{x}$:

$$S(\boldsymbol{x}) = \frac{\partial y}{\partial \boldsymbol{x}}. \tag{6}$$

We generalize this formulation to intermediate representations by considering the activations at layer $F_c(\boldsymbol{x})$, i.e.,

$$S(\boldsymbol{x}) = \frac{\partial F_c}{\partial \boldsymbol{x}}, \tag{7}$$

where $F_c(\boldsymbol{x}) = f_c \circ f_{c-1} \circ \cdots \circ f_1(\boldsymbol{x})$ denotes the composition of the first $c$ layers of the network.

### B.2 SmoothGrad

SmoothGrad extends gradient saliency by reducing noise through averaging over $N$ noisy perturbations of the input (Smilkov et al., 2017). Applied to the $F_c$ layer, the saliency map is given by

$$S(\boldsymbol{x}) = \frac{1}{N} \sum_{i=1}^{N} \frac{\partial F_c}{\partial \boldsymbol{x}_i}, \qquad \boldsymbol{x}_i = \boldsymbol{x} + \mathcal{N}(0, \sigma^2). \tag{8}$$

In our experiments, we used $N = 20$ samples and $\sigma = 0.1$.

### B.3 Perturbation-based saliency

Perturbation-based methods estimate the importance of an input feature $p$ by measuring the change in the output when that feature is altered, suppressed, or masked (Zeiler & Fergus, 2014; Greydanus et al., 2018; Fong & Vedaldi, 2017). Formally:

$$S(\mathbf{x})_p = y(\mathbf{x}) - y(\mathbf{x}_{[p]}),$$

where $\mathbf{x}_{[p]}$ denotes the perturbed version of $\mathbf{x}$ with feature $p$ modified. We adapted the method from Greydanus et al. (2018) originally applied to the output of the network (either to the policy estimate $\pi$ or the value estimate $V$) to the $F_c$ layer. Given an input frame $I_t$, the importance of pixel $(i, j)$ is estimated by perturbing the image with a Gaussian blur $A(I_t, \sigma_A)$ localized around $(i, j)$:

$$\Phi(I_t, i, j) = I_t \odot (1 - M(i, j)) + A(I_t, \sigma_A) \odot M(i, j), \tag{9}$$

where $\odot$ is the Hadamard product, and $M(i, j)$ is a Gaussian mask centered at $(i, j)$ with variance $\sigma^2 = 25$.

The saliency of pixel $(i, j)$ with respect to activations of the $F_c$ layer is then

$$S_\pi(t, i, j) = \frac{1}{2} \left\| F_{c,u}(I_{1:t}) - F_{c,u}(I'_{1:t}) \right\|^2, \tag{10}$$

where

$$I'_{1:k} = \begin{cases} \Phi(I_k, i, j), & \text{if } k = t, \\ I_k, & \text{otherwise.} \end{cases}$$

### B.4 Adaptation of the $h$-profile to saliency maps

To compare saliency methods, we adapted the $h$-profile to operate directly on saliency maps instead of LRP relevance maps. The adapted version of $h$ noted $\tilde{h}$, is defined as

$$\tilde{h}(o_g) = \frac{1}{|\boldsymbol{X}|} \sum_{\boldsymbol{x} \in \boldsymbol{X}} \bar{S}_g(\boldsymbol{x}), \tag{11}$$

with

$$\bar{S}_g(\boldsymbol{x}) = \sum_{p \in P_{o_g}} S_p(\boldsymbol{x})$$

where $o_g \in O = \{o_1, o_2, \ldots, o_G\}$ is an object in the input space, and $P_{o_g}$ is the set of pixels corresponding to $o_g$ according to the mapping $\bar{\boldsymbol{X}}$.

The resulting $\tilde{h}$-profile provides an average saliency measure per object, enabling a direct comparison of different saliency methods with the LRP-based $h$-profile.

## C Dataset and Labeling

We considered two dataset generation strategies:: 1. Online datasets, constructed individually at the time of analysis. 2. Constant datasets, generated once per environment and reused across agents and training runs.

Labeling Procedures:

1. Atari 2600: For moving objects, labels were extracted from RAM coordinates (with the help of the annotated ram in Anand et al. (2019)); for static objects, labels were assigned by fixed position.

2. Custom Pong: Labels were generated automatically from the color map, using our source code.

3. Biomechanical Tasks (MuJoCo): Labels were obtained through MuJoCo's built-in automatic labeling.

Dataset Construction: For automatic generation, agents interacted with the environment until the minimum number of samples was reached. A four-frame input was included only if (i) all objects were present, (ii) objects did not overlap, and (iii) the input was unique.

For constant datasets, we generated labeled data from 10 untrained and 10 trained A2C agents (distinct seeds). From each agent, three inputs were sampled, yielding 60 input–label pairs, which were then subsampled to 50.

Constant datasets offer three advantages: 1.Ensure comparability across learning, agents and algorithms (e.g. in the biomechanical environment the input features (joint positions, velocities, accelerations) vary substantially over training, potentially biasing attention analyses. Fixing the dataset eliminated this source of variability). 2. Mitigate failures in dataset generation when agents converge to suboptimal policies (e.g., in Space Invaders when the agent fails to shoot, leaving objects absent). 3.Reduce computational overhead by avoiding repeated dataset construction.

We compared analyses with and without constant datasets in Breakout to assess if this choice has a significant impact on the results (Fig. 6). We found that in the case of the Breakout game, the overall results are consistent for both type of dataset (constant or not). The main difference can be found for DQN and QRDQN agents where the amplitude of the attention on the ball and on the bricks are different around 1M steps and 4M steps respectively. However, the overall dynamic, the final attention profile at the end of the training and the overall dependence on the bricks feature is present in both cases (constant and not constant dataset) for both QRDQN and DQN agents.

## D  Sensitivity Analysis

The hierachical-attention profile $h$ rely on two main hyperparameters. The first one is $q$ and is related to the filtering steps over the $F_c$ neurons. To speed up the computation of $h$ and reduce its noise, we only computed the relevance map for the top neurons making up $q\%$ of the relevance for the network output. The lower the $q\%$ the less neurons we take into account and vice versa. The second hyperparameter is the number of input images $N$ on which to evaluate the agent's attention at each measurement. A game with a higher number of states would require more input images to adequately cover the different game states.

We performed a Sobol anaylsis with the python library SALib (Sensitivity Analysis Library) Herman & Usher (2017); Iwanaga et al. (2022) to measure the sensitivity of $h$ outputs and computation time to the parameters $q$ and $N$ (number of images) for an agent in each game and game version. The sampler generated 3072 samples of the $q$ and $N$. The q-values ranged between 0.1 and 1 and the number of images ranged between 5 and 200. Results of the analysis for the Atari 2600, the Custom Pong and the Biomechanical environments are displayed Figs. 8 to 11.

We found that although the number of images $N$ had an impact on the values of $h$, the overall $h$ patterns was consistent even for small values of $N$. However, the computation times increases with the number of images. For these reasons we chose to create dataset with $N = 50$.

## E  Network's architecture

**Atari 2600 and Custom Pong**   For Atari 2600 and custom Pong experiments, all agents shared the same convolutional architecture. The network consisted of three convolutional layers (sizes: $4 \times 32$, $32 \times 64$, and $64 \times 64$), followed by a fully connected layer of 512 units (denoted as $F_c$). The final output layer depended

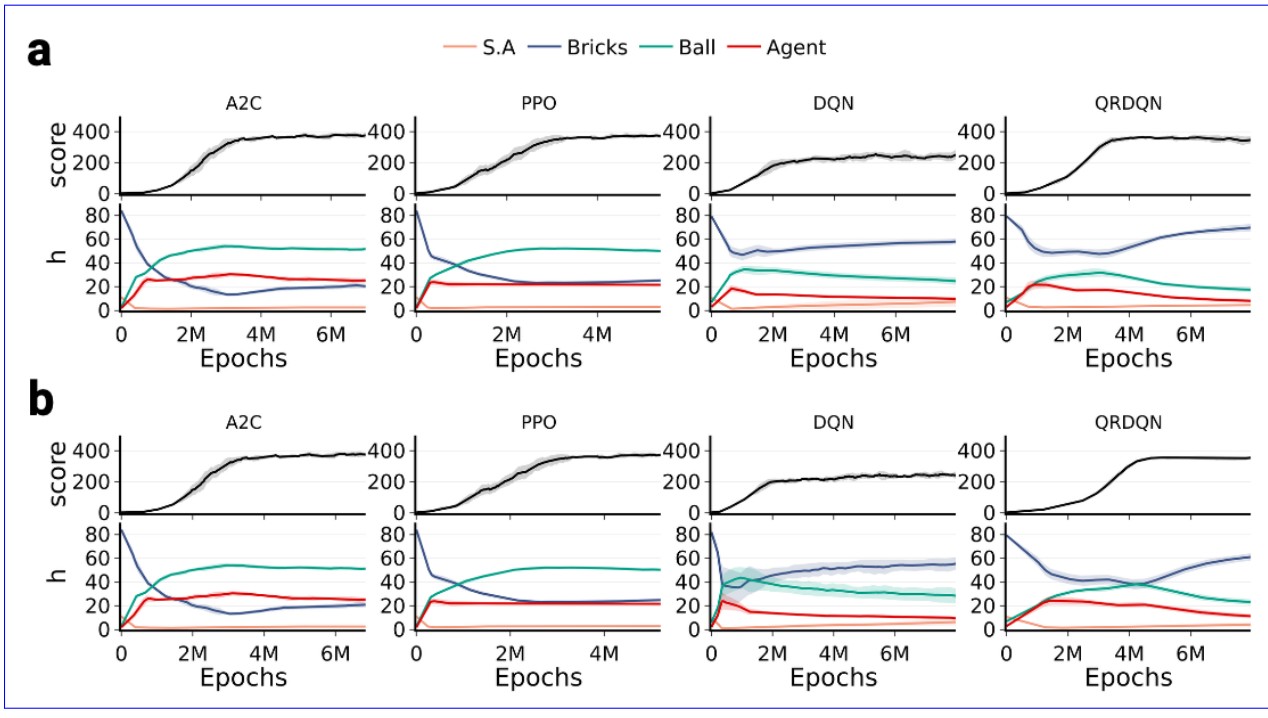

Figure 6: Breakout training curves showing reward progression (top) and h-profile trajectories (bottom) computed with **a** a constant dataset and **b** datasets generated at the time of the analysis, for PPO, A2C, DQN, and QR-DQN

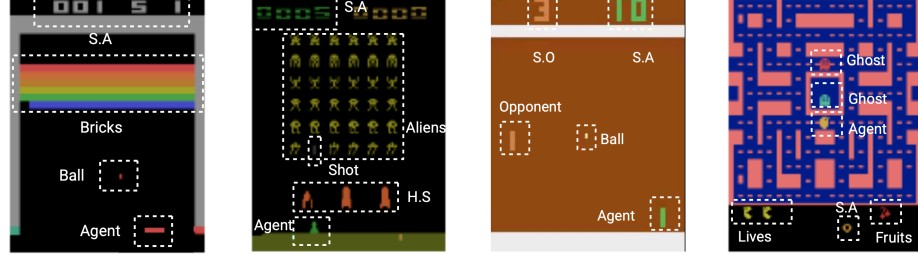

Figure 7: **Objects tracked in Atari 2600 attention analysis**. Set of annotated objects used for hierarchical-attention computation in four Atari 2600 games (Ms Pacman, Space Invaders, Breakout, Pong).

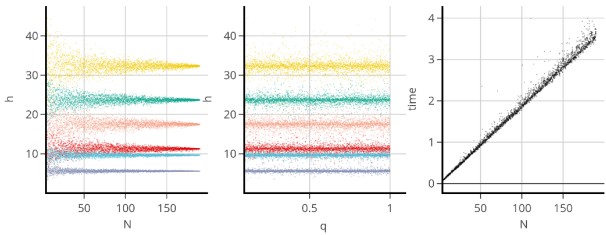

Figure 8: Impact of the number of images ($N$) and of the $q$ value on $h$ for an agent trained on version $v2$ of the Custom Pong game

Figure 9: Impact of the number of images ($N$) and of the $q$ value on $h$ for an A2C a PPO agent trained on **a** Pong, **b** Breakout, **c** Space Invaders, and **d** Ms Pacman

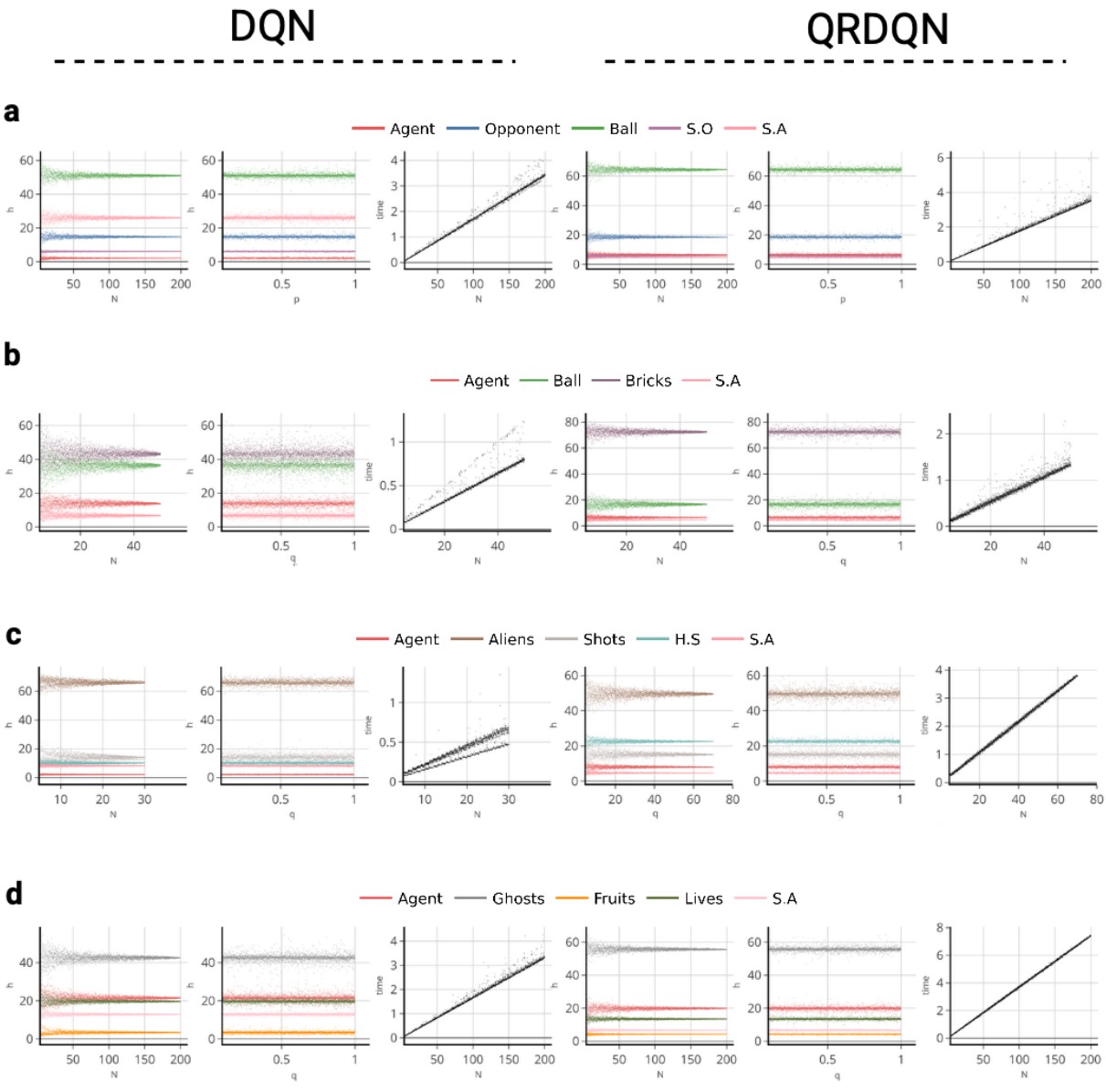

Figure 10: Impact of the number of images ($N$) and of the $q$ value on $h$ for an DQN a QR-DQN agent trained on **a** Pong, **b** Breakout, **c** Space Invaders, and **d** Ms Pacman

on the learning algorithm: a value head for DQN and QR-DQN, or combined policy and value heads for A2C and PPO.

**Biomechanical Tasks (User-in-the-Box)** For biomechanical control tasks, PPO agents used a multi-input policy architecture. Observations from different modalities were first encoded separately before being combined in a shared actor–critic network:

- **Visual encoder:** a three-layer CNN followed by a fully connected layer, processing 1–4 visual input channels depending on the modality.

- **Non-visual encoder:** a fully connected layer with 128 units and Leaky ReLU activation, processing proprioceptive and other non-visual inputs.

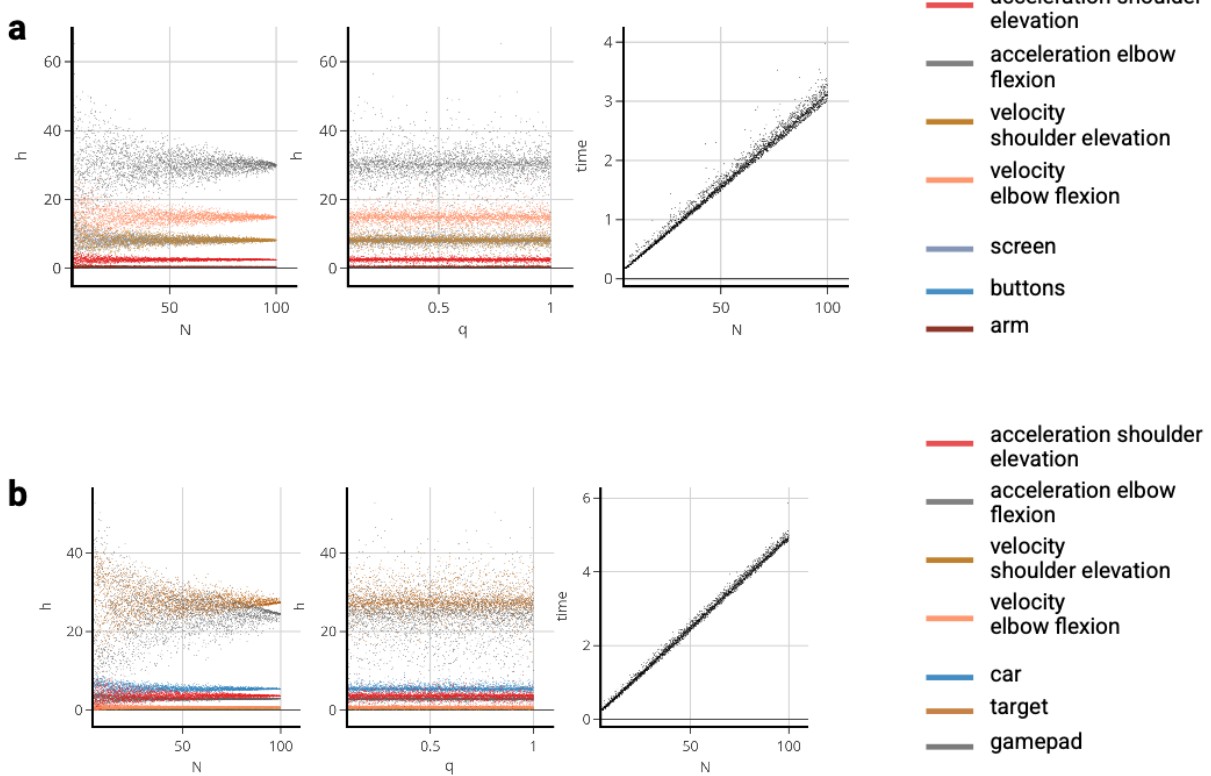

Figure 11: Impact of the number of images ($N$) and of the $q$ value on $h$ for a PPO agent on **a** Choice reaction task and **b** Parking a remote control car task.

The encoded representations were concatenated and passed through two shared fully connected layers (256 units each, Leaky ReLU). The actor head produced 26 tanh-activated outputs corresponding to muscle activations, while the critic head produced a single scalar value.

## F   RL training procedure

For A2C agents training on Atari 2600 environments and on the Custom Pong games we set, the learning rate to $7e^{-4}$, the discount factor 0.99, the entropy coefficient 0.01 and the value loss coefficient 0.25. The number of parallel environment was set to 100.

For PPO agents training on Atari 2600 environments we set, the learning rate to $2.5e^{-4}$, the discount factor to 0.98, the entropy coefficient to 0.01 and the value loss coefficient to 0.5.The number of parallel environment was set to 100.

For PPO agents training on the parking a remote control car task the learning rate follows a linear schedule with initial value of $5e^{-5}$, min value of $1e^{-7}$ and a threshold of 0.9. The batch size was set to 1000 and the number of environment was set to 20.

For DQN agents we set, the learning rate to $1e^{-4}$, the discount factor to 0.99, the buffer size to 100000, the batch size to 32, the train frequency to 4, the target update interval to 1000, the exploration fraction to 0.1 and final exploration to 0.01 .

For QR-DQN agents we set the number of quantile to 50, the learning rate to $5e - 5$, , the discount factor to 0.99, the buffer size to 100000, the batch size to 64, the train frequency to 4, the target update interval to 10000, the exploration fraction to 0.025 and final exploration to 0.01.

Training was carried out on Nvidia via A100 GPUs using a single GPU with up to 18 CPU cores per task and a memory of 125 GB.

## G   Measurements of the attention during training

The frequency of evaluation $f_r$, which indicates how often the $h-$profile was extracted, was chosen for each game according to the observed training dynamics. Table 2 summarize the the choice of $f_r$ for each environment and DRL model. The choice of $f_r$ for each environment was influenced by the number of steps required for the network to demonstrate an increase in the training score. It is important to note that different frequencies could be selected based on the desired granularity for studying the evolution of the profile $h$. Our selected frequencies aimed to balance the need for detailed profile $h$ tracking with the practical constraints of computational resources and training time.

Table 2: Frequency analysis

|  | A2C | PPO | DQN | QR-DQN |
|---|---|---|---|---|
| Pong | 100 | 40 | 10000 | 10000 |
| Breakout | 300 | 200 | 50000 | 50000 |
| Space Invaders | 100 | 100 | 10000 | 10000 |
| Ms Pacman | 250 | 100 | 10000 | 10000 |
| Custom Pong $v_0$ | 230 | | | |
| Custom Pong $v_1$ | 85 | | | |
| Custom Pong $v_2$ | 50 | | | |
| Parking a remote control car | | 4000 | | |
| Choice reaction test | | 2000 | | |

## H   Game

### H.1   Game Dynamics Setup

We developed a modified version of the classic Atari 2600 Pong game using the Pygame library (v.2.5.2 Shinners (2011)) to create a custom environment for our experiments. This variation maintains the original game's elements—paddles, walls, and scoring—but introduces dual balls instead of one. The dimensions and colours of the game components mirror those in the original Pong. In this setup, each paddle is distinctively colored, as are the two balls. The paddles move on the y-axis at a rate of 2 pixels per frame, while the balls move at speeds of 4 pixels per frame along the x-axis and 2 pixels per frame on the y-axis. Game states are represented as 84x84 pixel RGB images.

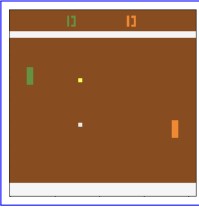

Figure 12: Example of a frame from the Custom Pong environment version $v2$. The agent paddle is in green.

### H.2   Preprocessing and Environment Wrapping

To prepare the game environment for reinforcement learning (RL), we encapsulated it within the same preprocessing wrapper used for Atari 2600 games in the Stable Baselines 3 framework (v.3 Raffin et al. (2021)). This wrapper converts each game frame from RGB to grayscale to streamline input dimensions and reduce computational demands (Mnih et al., 2013). To prevent the RL agent from memorising action sequences, we introduced randomness in the initial ball direction. Last but not the least, the observation input given to the agent is not Markovian as all the information necessary to predict the next input given this input and an action is not available. Aligning with prior research by Y and common practices in Stable Baselines 3, we stacked four consecutive frames, producing a composite input $o_t \in R^{4 \times 84 \times 84}$, to provide the agent with a temporal context for decision-making.

### H.3   Variations

The game variations are summarised in Table Y.

Table 3: Design of the game variations.

|      | B1  | B2  | B1 D | B2 D | B1 R | B2 R |
|------|-----|-----|------|------|------|------|
| V0   | Yes | No  | D1   | x    | R1   | x    |
| V1   | Yes | Yes | D1   | D1   | R1   | x    |
| V2   | Yes | Yes | D1   | D2   | R1   | R2   |

**Dynamics:**

- D1: The ball rebounds off both the walls and the paddles.

- D2: The ball is capable of bouncing off the walls and the agent's paddle. However, it will not rebound off the opponent's paddle, and pass through it.

**Rewards:**

- R1: The agent is awarded +1 for scoring a point on the opponent's side, and receives a -1 if the agent fails to hit the ball, allowing it to pass by.

- R2: The agent gains +1 for successfully hitting the ball with their paddle. Conversely, a -1 penalty is applied if the agent fails to hit the ball, allowing it to pass by.

The balls start in the middle of the screen. They have the same x-direction that is randomly generated and opposite y-directions. This choice of initial states forces the agent to choose between the balls for the first hit. The condition for an episode to end and for the balls to respawn is dependent on *B1* being scored. The opponent is hard-coded to position its y-axis on the y-axis of *B1*.

## I   Atari 2600

### I.1   Dissimilarity of the h-profile between algorithms

Fig. 13 details the *h*-profile dissimilarity at the algorithm-pair level.

### I.2   Attention development during training Atari 2600

Similarly to the Breakout experiments, we computed the attention profile $h$ during training alongside the performance score for PPO, A2C, DQN, and QR-DQN agents on Pong, Space Invaders, and Ms. Pacman. For each game–algorithm pair, we trained ten agents initialized with different random seeds. The set of

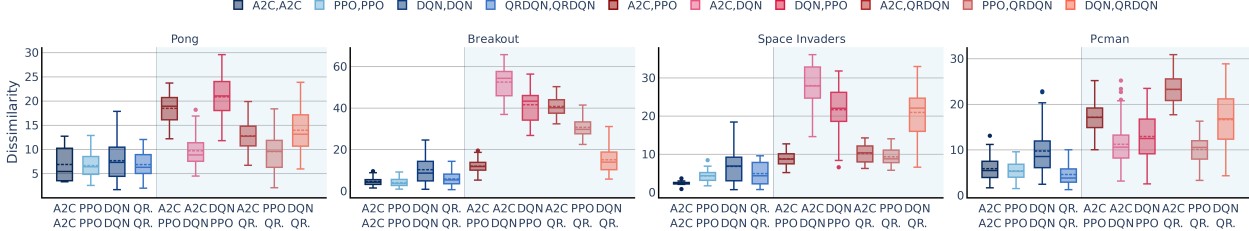

Figure 13: Pairwise dissimilarity (Euclidean distance) between hierarchical-attention profiles at the end of training within and between learning algorithms. Detals for each pair of algorithm.

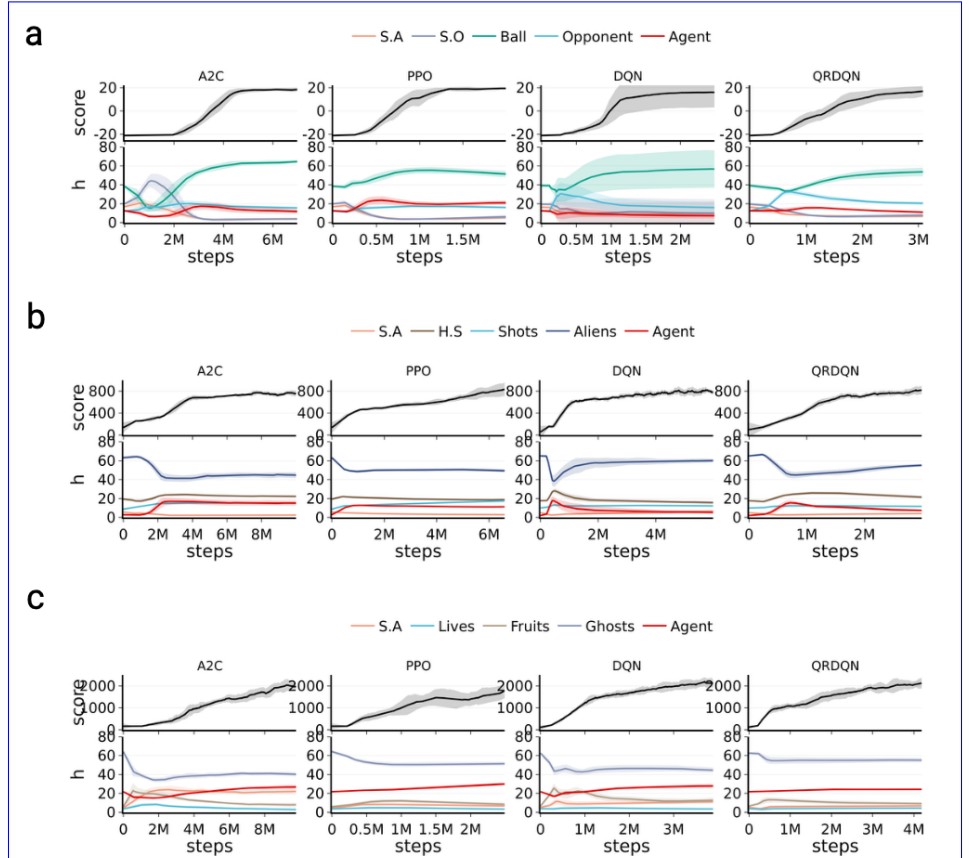

Figure 14: **a** Pong, **b** Space Invaders, **c** Ms. Pacman training curves showing reward progression (top) and h-profile trajectories (bottom) for PPO, A2C, DQN, and QR-DQN.

objects defined for each game is illustrated in Fig. 7, and the analysis frequency is reported in Table 2. The results are illustrated Fig. 14

Consistent with the observations made for Breakout, we found that the majority of changes in attention occur early in training. While some algorithm-specific differences were present (e.g., in Pong, A2C agents showed a stronger early focus on the opponent's score compared to PPO, DQN, and QR-DQN), the overall dynamics of attention during the early training phase were broadly similar across algorithms.

### I.3    Impact of the buffer size on the attention profile

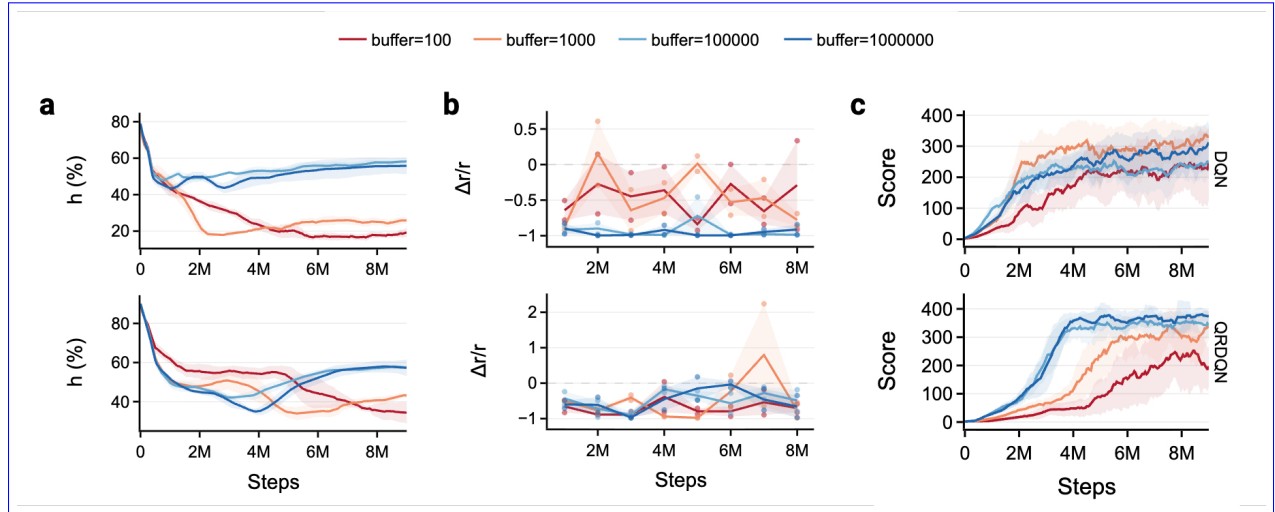

Figure 15: **Buffer size impacts the attention to the bricks** Attention to the bricks during training (left column) and robustness under perturbations of the bricks (right column) for different buffer sizes for DQN and QR-DQN. Average over three agents is represented in bold line, shaded areas represent one standard deviation.

To investigate the divergence in attention profiles between value-based methods (DQN, QR-DQN) and policy-gradient methods (A2C, PPO) in Breakout, we examine the role of experience replay. A key difference between these approaches is the use of a replay buffer in DQN and QR-DQN, which stores past transitions and allows them to be reused multiple times during training. This mechanism can amplify the influence of rare but informative events.

We hypothesize that replay buffers bias the learned attention toward object configurations associated with such events (e.g., specific brick patterns). To test this hypothesis, we train DQN and QR-DQN agents with varying replay buffer sizes ($10^2, 10^3, 10^5, 10^6$). For each configuration, we measure (i) the attention allocated to bricks, (ii) task performance, and (iii) robustness to visual perturbations of the bricks.

As shown in Fig. 15 **a**, the size of the replay buffer strongly affects the learned attention profile. Larger buffers ($10^5, 10^6$) lead to substantially higher attention to bricks (e.g., increasing from ~25% to ~60%), whereas smaller buffers produce attention levels closer to those observed in A2C and PPO. This trend is consistent with the fact that smaller buffers induce a training distribution closer to on-policy data. We also observe a corresponding effect on robustness (Fig. 15 **b**): DQN agents trained with smaller buffers are more robust to brick perturbations, aligning with their reduced reliance on brick-specific features. For a similar performance score (Fig. 15 **c**), a judicious choice of buffer size can therefore improve the resistance to perturbation. For a comparable level of performance (Fig. 15 **c**), the choice of buffer size therefore influences robustness to perturbations. For QR-DQN, this effect was less pronounced, suggesting that additional factors may contribute to its brick sensitivity.

## J  Custom Pong

### J.1  Color swap between B1 and B2

In this experiment, we investigated whether the color assigned to each ball affected the $h-$profile. We conducted this experiment by training 20 models for each game version, exchanging only the colors of balls B1 and B2. As observed on Section J.1, the color of the ball does not impact the overall hierarchical-attention pattern with agents trained on *v1* focusing more on B1 and agents trained on *v2* focusing more on B2.

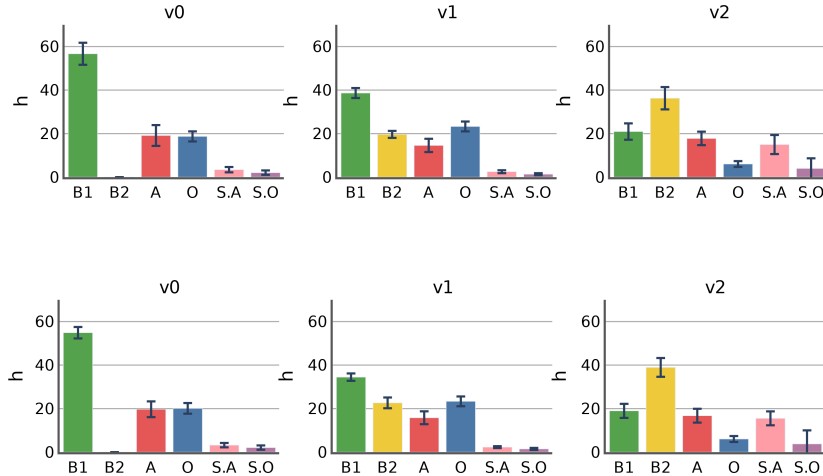

Figure 16: **(top)** Hierarchical attention average across 50 agents with an initial color mapping: B1 color (236, 236, 236) and B2 color (255, 255, 0). **(bottom)** Hierarchical attention average across 20 agents with a swapped color mapping: B1 color (255, 255, 0) and B2 color (236, 236, 236). Error bars represent the standard deviation.

## K  User-in-the-box

### K.1  Fine-grained Attention During Training

For a more detailed analysis of the biomechanical tasks, we redefined the set of annotated objects used to compute the attention profile $h$. Instead of tracking attention only at the modality level (vision vs. proprioception), we examined finer-grained inputs. For proprioception, we defined each proprioception input as an objet in both tasks. For visual input, we tracked task-relevant objects: the car, the target, and the gamepad in the Parking a remote control car task, and the screen, buttons, and arm in the Choice reaction task.

Fig. 17 shows how attention toward a subset of these objects evolved during training, alongside performance and behavior.

Across both tasks, attention to proprioceptive accelerations decreased over training, while attention to velocities increased. This shift is consistent with the temporal demands of the tasks, where controlling movement speed becomes critical for maximizing reward. In parallel, attention to visual cues also increased as training progressed, reflecting their growing importance for accurate task completion (beyond reaching a fixed gamepad or fixed buttons as explained in Section 4.3 of the main text).

### K.2  Moving environment

Fig. 18 displays the details of the attention distribution on the buttons for agents trained on the non moving buttons environment and the moving buttons environment of the Choice reaction task.

## L  Experiments with other saliency maps

### L.1  Comparison on objects focus

As out study is primarily focused on the attention given to objects we measured how much relevance was given to the objects (as opposition to the background) for each saliency method. We measure the attention

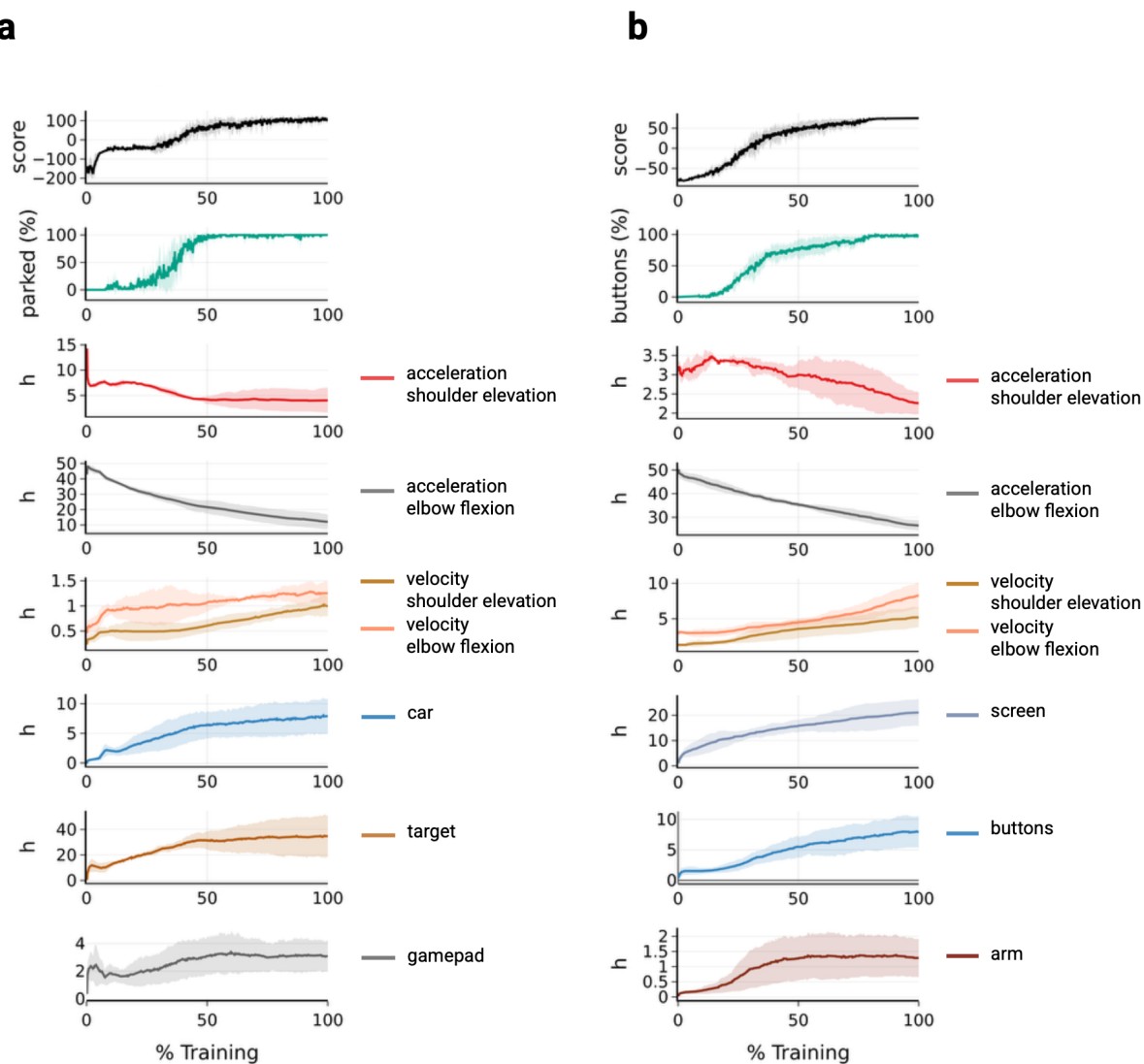

Figure 17: Evolution of performance, behavior, and attention allocation in the Parking a remote control car task **a** and the Choice reaction task **b**. Curves represent the average across three agents; shaded regions indicate standard deviation. Attention is shown for individual proprioceptive features and visual objects.

profile including the background (that is to say all pixels not belonging to the object defined) to 50 agents trained in each version of the Custom Pong game. Fig. 19 shows that on average, LRP method is the method distributing most of its relevance on the objects compared to grad, smoothgrad and perturbation that allocates more than 60% of the relevance on the background of the game. A more detailed representation of the saliency maps given by each method is shown in Table 4 with the Pong game from Atari 2600.

Overall, the saliency maps obtained using LRP are less noisy (Table 4) with higher relevance given to objects (Fig. 19 ) which validates out choice to used LRP as the main saliency method to compute $h$.

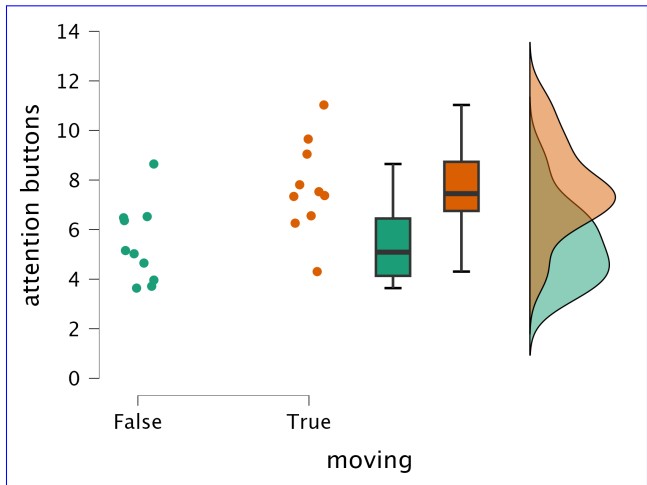

Figure 18: Attention on the buttons for 10 agents trained on the static task (moving False) and 10 agents trained on the moving task (moving True)

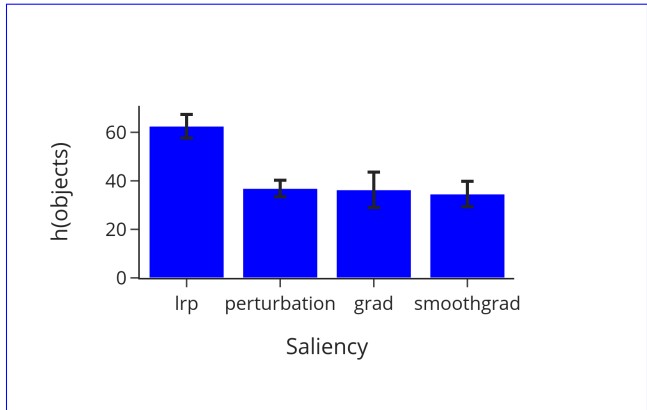

Figure 19: Attention profile on the objects averaged over all trained agents agents and Custom Pong game version (50 agents per version), for each saliency method. The error bar represents the standard deviation.

### L.2 Comparison on the Atari 2600 Games

#### L.2.1 Attention profile dissimilarity between algorithm

We extended the analysis from Section 4.1 to the three other saliency methods by training five agents per algorithm and game for each method. The results are shown in Fig. 20, which reports both the performance scores (left column) and the dissimilarity of attention profiles (right column) for Breakout (Fig. 20**a**) and Space Invaders (Fig. 20**b**). These extended results confirm the observation with the LRP method: attention profiles differ substantially across algorithms, even when task performance is comparable.

#### L.2.2 Attention dynamics

We reproduced the measurement of attention profiles for five agents per algorithm across training using each saliency method (Fig. 21). In general, the temporal dynamics were consistent across methods, although the absolute importance assigned to individual objects varied. For Breakout, the strong preference of DQN and QR-DQN agents for the bricks feature was consistently observed across all saliency methods. By contrast, in PPO and A2C agents, SmoothGrad appeared to amplify the importance of the bricks relative to the other three saliency maps (LRP, gradient, and perturbation). This effect may stem from the noisier nature of

Table 4: Comparison saliency maps for Pong

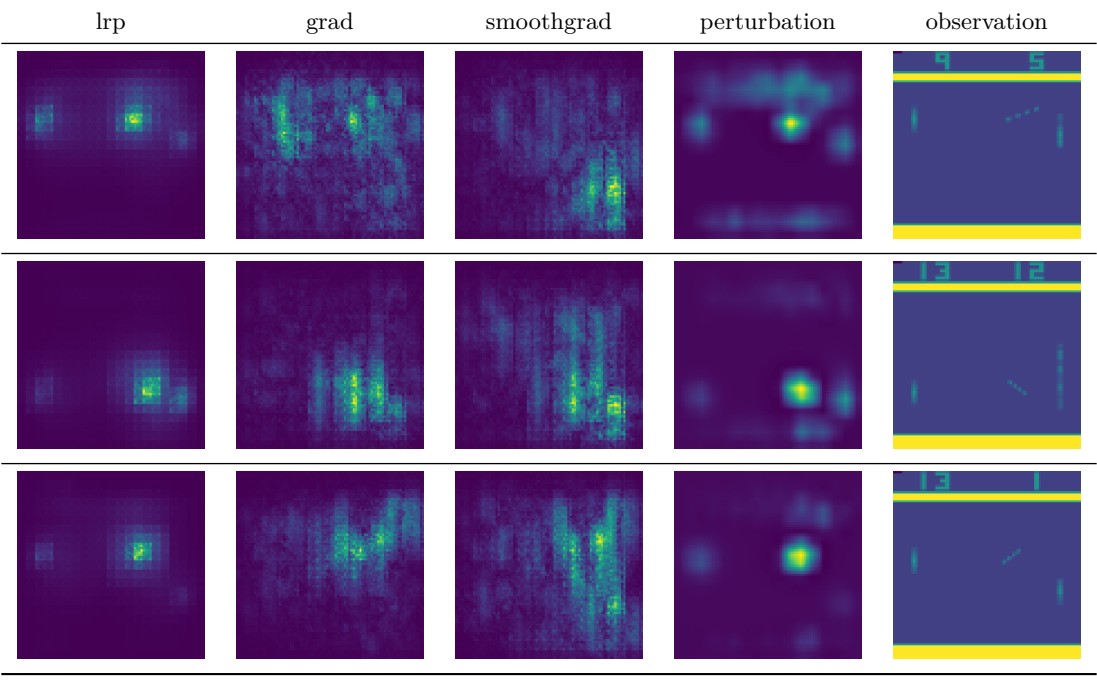

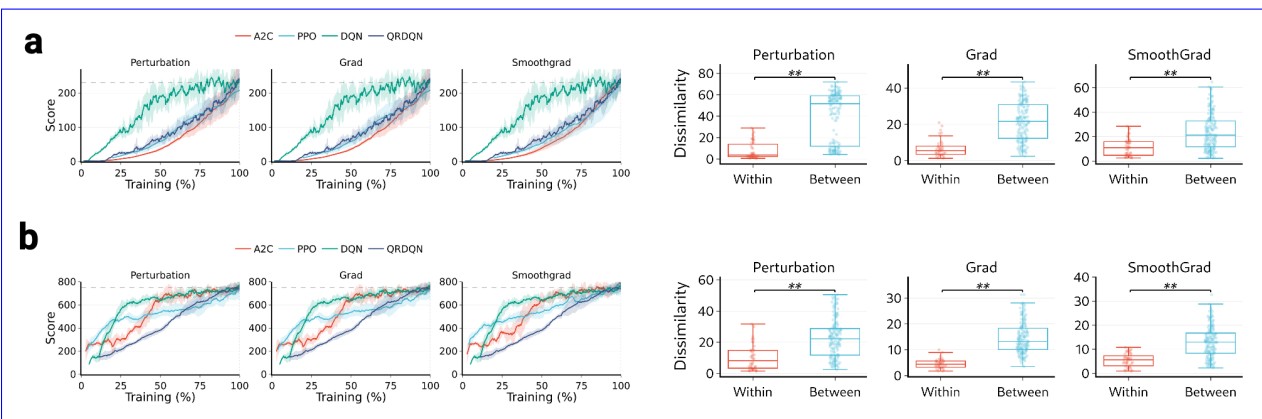

Figure 20: From left to right: score averaged over 5 agents per learning algorithm during training and ANOSIM analysis of the $h$-profile at the end of the training within and between learning algorithm for the Breakout game (**a**) and the Space Invaders game (**b**) for each saliency maps.

SmoothGrad, where salience attributed to the ball or score display can spread into the bricks region and become further amplified by the averaging process inherent to the method. These observations highlight the value of complementary analyses—such as perturbation-based experiments—to rigorously evaluate and validate hypotheses generated by saliency methods. Fig. 22 displays the comparison of the $h$-profile during training on the Space-Invaders game. Here as well, the overall dynamic is conserved although different absolute values can be found between methods.

### L.3 Comparison on the Custom Pong Games

The attention profile of 50 agents trained on versions $v_0$, $v_1$ and $v_2$ was computed for each saliency maps and displayed

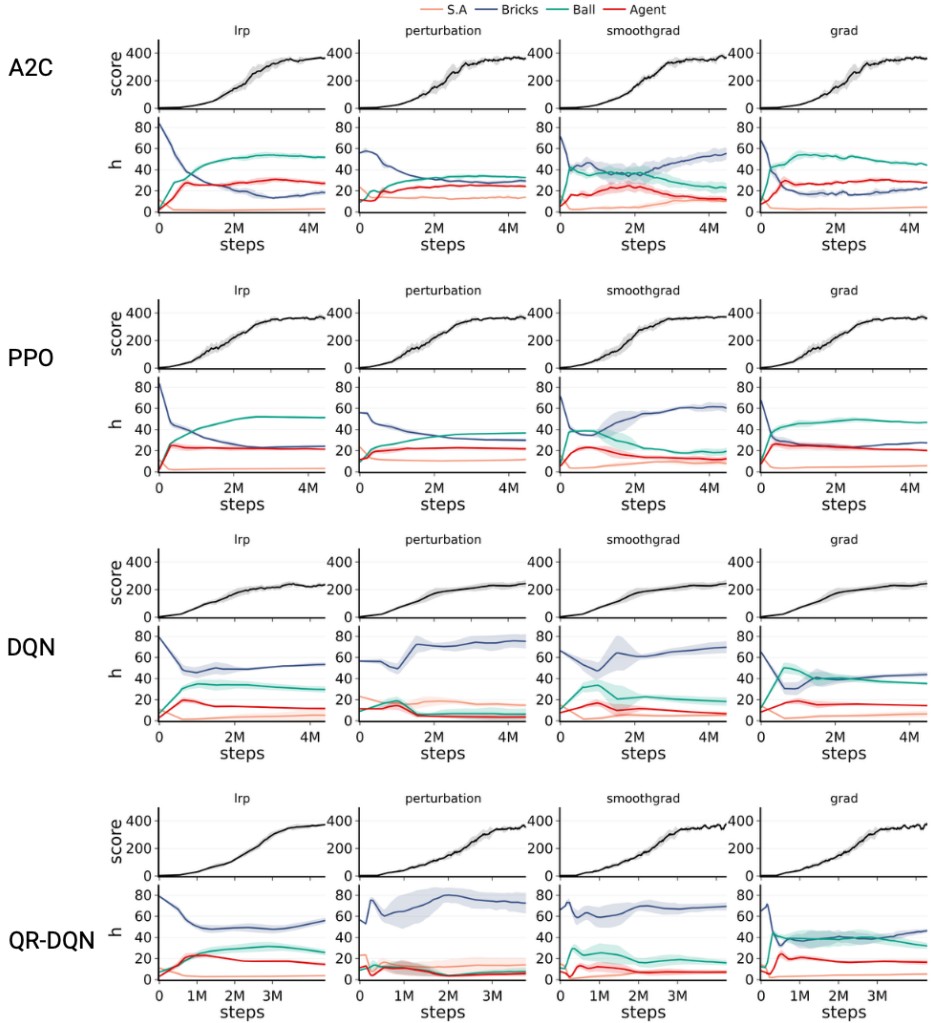

Figure 21: Breakout training curves showing reward progression (top) and h-profile trajectories (bottom) for PPO, A2C, DQN, and QR-DQN computed with each saliency maps.

The ball preference for each game version is preserved across saliency method, with an overall preference for $B_1$ in $v_1$ and $B_2$ in $v_2$. However, the difference is more distinctive in the LRP method. A major difference between the saliency methods is a stronger attention to the score of the agent (S.A) for the perturbation, grad and smoothgrad methods compared to the LRP methods. A potential interpretation for this difference is that as these methods do not have sharp saliency on the object (Fig. 19 and Table 4), if the agent gets closer to the top wall where its score is displayed, the saliency on the agent might get attributed to its displayed score (S.A object).

## L.4   Comparison on the Biomechanical model

Due to the long training time (several days) of agents trained on both biomechanical task and the particular case of multimodal inputs and regression output we limited the reproduction of the experiments to the Gradient saliency map (which is the building block of the usual Layer-Wise-Relevance propagation method already used) and to the impact of the moving buttons on the agent's attention in the Choice reaction task. Fig. 24 shows that agent's attention toward the buttons is on average higher for agents trained on the task

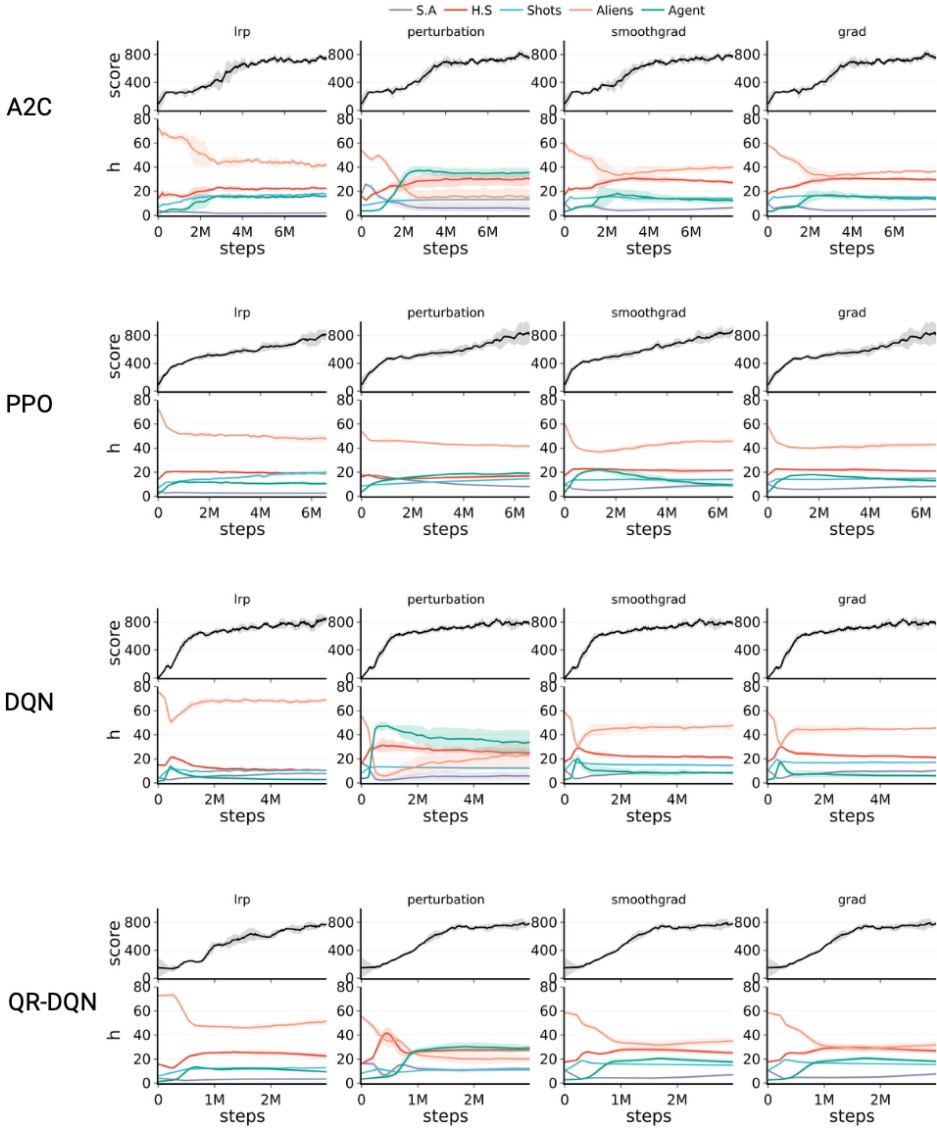

Figure 22: Space Invaders training curves showing reward progression (top) and h-profile trajectories (bottom) for PPO, A2C, DQN, and QR-DQN computed with each saliency maps.

with moving buttons, confirming the findings with the LRP methods. However, the t-statistical test was not conclusive with $p > 0.01$.

## M  Complementary experiments

### M.1  Sensitivity of the h-profile to object's granularity and labeling noise

To evaluate the sensitivity of $h$-profiles to object granularity and labeling noise, we conducted a controlled perturbation study in the remote driving environment. This MuJoCo environment naturally allows multiple levels of visual segmentation, enabling the same observations to be annotated at different object granularities. We defined four levels of segmentation:

- Level 1 (coarse): simulated user, car, target, and controller.

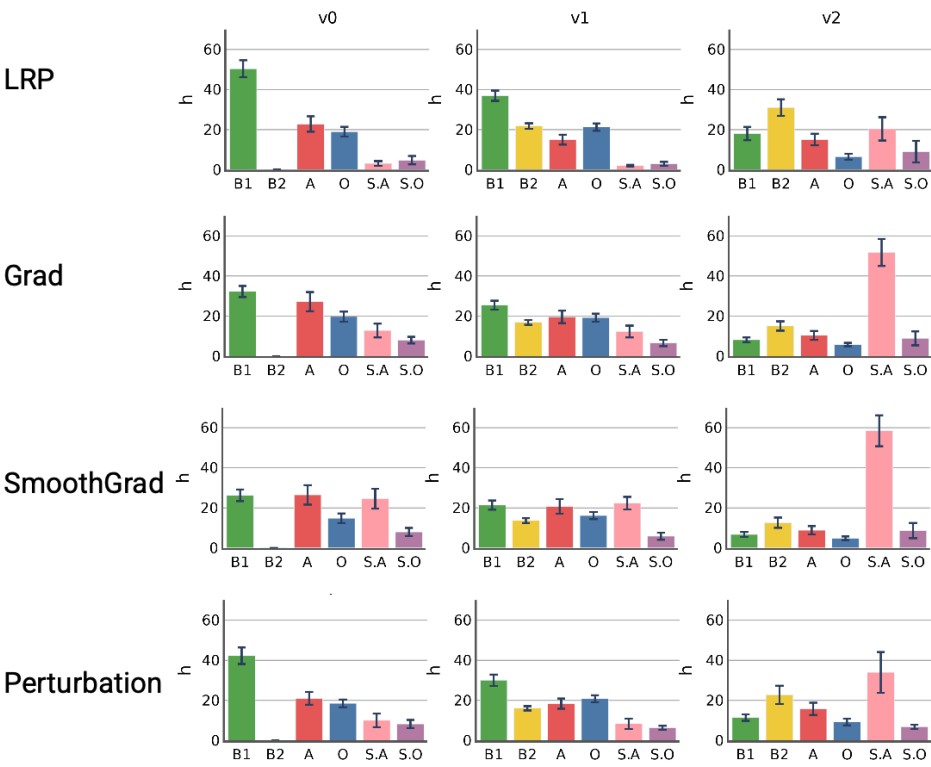

Figure 23: Comparison saliency maps for Custom Pong

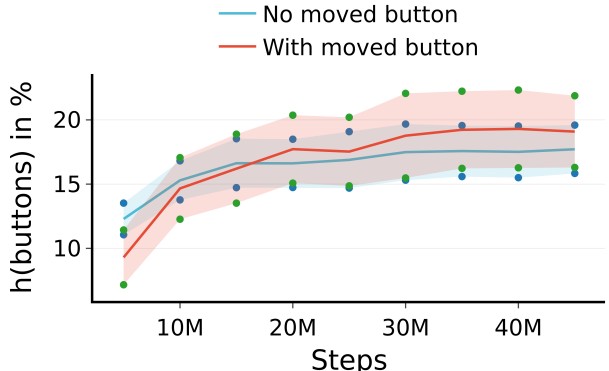

Figure 24: Increased attention towards the button objects in the Choice reaction task for agents trained on the environment with moving buttons. The average is over 10 agents and the standard deviation is represented with a shaded area.

- Level 2: muscles, skeleton, car, target, and controller.

- Level 3 (fine semantic parts): finger, forearm, hand, shoulder girdle, tendon, thumb, upper arm, axle of the car, main body of the car, wheels, controller buttons, controller body, thumbstick, target, and ground.

- Level 4 (geometry-level): all individual MuJoCo geometries composing the visual observation.

To simulate annotation imperfections, we introduced three types of object-level labeling noise:

- Relabeling: an object's label is replaced with another label with probability $p$.

- Object removal: an object is reassigned to the background with probability $p$.

- Boundary dilation: the spatial extent of an object label is expanded with probability $p$.

Using a dataset of 100 observations annotated at each granularity level, we computed $h$-profiles for both clean and perturbed labels across multiple values of $p$ for a single trained agent. For each perturbation level, three random noise iterations were generated. Sensitivity was quantified as the Jensen–Shannon distance between the $h$-profiles obtained from clean and perturbed annotations, averaged across noise seeds.

Figure 25 shows the attention distribution for each granularity level. At the finest level, however, interpretation should be treated cautiously, as the resulting profile may be limited by the spatial resolution of the saliency map.

Figure 26 illustrates how robustness to labeling noise depends on object granularity. Coarser object definitions appear more sensitive to object removal, whereas the finest granularity (level 4) shows higher sensitivity to relabeling. This pattern likely arises because coarse segmentations concentrate attention on a small number of objects, making the removal of one object particularly impactful, whereas finer segmentations distribute attention across a larger number of smaller components. As above, interpretation at the finest granularity should be treated cautiously due to the spatial resolution limits of the saliency maps.

## M.2   Impact of object size and speed on the attention profile

We analyzed how variations in object size and speed affect the agent's attention profile. To this end, we trained A2C agents on multiple variants of the Custom Pong environment in which either the size of the game objects (ball and paddles) or their velocities were modified.

Since changes to the environment may affect learnability, we restricted our analysis to configurations in which agents successfully learn the task. For each condition and random seed, we collected a dataset of 50 observations using the initialized policy network and computed the corresponding attention profiles.

Figures Figs. 27 and 28 report reward progression (top) and attention profiles (bottom) for different configurations of size and speed respectively. We distinguish between two sources of variation in these profiles: (i) changes induced by the attention estimation method (e.g., sensitivity to object size), and (ii) genuine shifts in the learned policy.

**Effect of object size.**  Altering object size affects both the visual observation and the learned behavior. For instance, when the ball size is increased while paddle sizes remain unchanged ($S_a = S_o = 1$, $S_b = 2$), the agent allocates substantially more attention to the ball at initialization compared to the reference setting. During training, attention to the paddles decreases, whereas it slightly increases in the reference environment. This suggests a shift in strategy in which the agent relies more heavily on tracking the ball position. This change is also reflected in the learning dynamics: the agent converges after approximately 2M steps in this condition, compared to about 3M steps in the reference setting and roughly 5M steps when all objects are uniformly reduced in size.

**Effect of object speed.**  We next modify object velocities while keeping sizes fixed, enabling a more controlled comparison since visual appearance remains unchanged. In this setting, the main differences arise in the attention allocated to the agent's paddle. In particular, when the ball speed is increased while paddle speed remains unchanged, the agent allocates substantially more attention to its own paddle, eventually exceeding the attention given to the opponent. This likely reflects increased task difficulty, as the faster ball reduces the margin for successful interception. Consistently, training also takes longer in this configuration (approximately 4M steps compared to around 3M steps in other conditions).

Overall, these results indicate that variations in object size and speed influence the learned attention profile. Part of this effect can be attributed to biases in the attention estimation method, while another part reflects genuine adaptations in the agent's strategy.

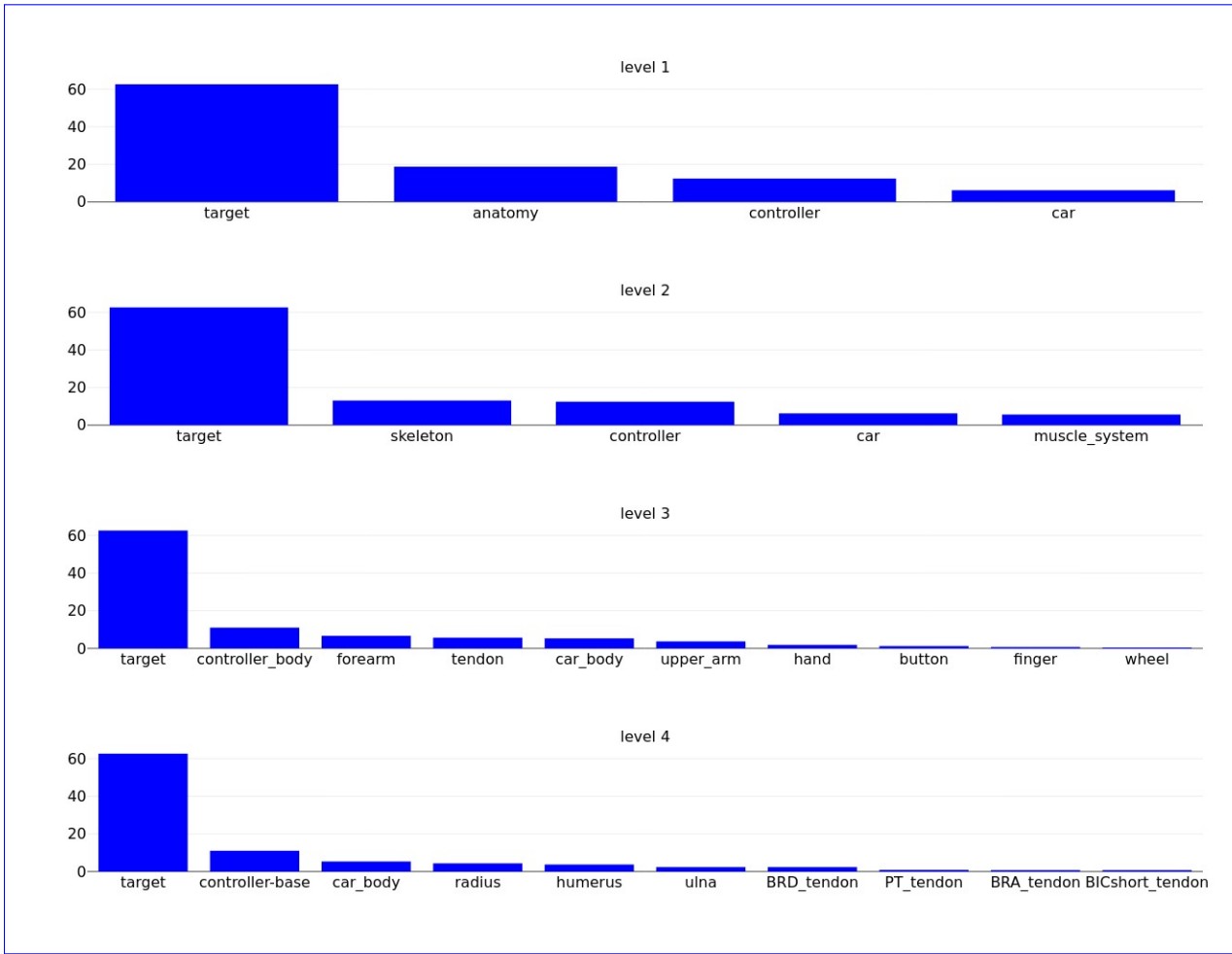

Figure 25: **Attention profiles across object granularities.** Average $h$-profiles for the four segmentation levels used in the remote driving environment. Each bar represents the proportion of attention assigned to a visual object. From top to bottom, the plots correspond to increasingly fine object granularities. For readability, only the 10 objects with the highest attention values are shown.

**Limitations.** Our analysis is subject to several limitations. First, attention profiles are computed using LRP, which may be sensitive to low-level visual properties such as object size, potentially confounding interpretation. Finally, changes in environmental parameters (size and speed) jointly affect both perception and task dynamics, making it difficult to fully disentangle measurement artifacts from genuine behavioral adaptations. A more controlled study in simplified environments would be required to isolate these factors.

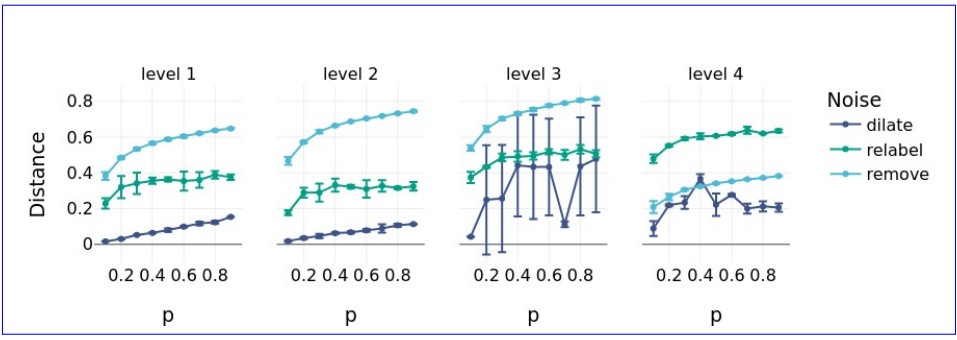

Figure 26: **Sensitivity of attention profiles to labeling noise.** Jensen–Shannon distance between the original $h$-profile and profiles computed from noisy annotations,as a function of the noise intensity $p$. Results are shown for three types of annotation noise (relabeling, object removal, and boundary dilation) across the four segmentation granularities. Distances are averaged over three noise iterations, and error bars represents one standard deviation.

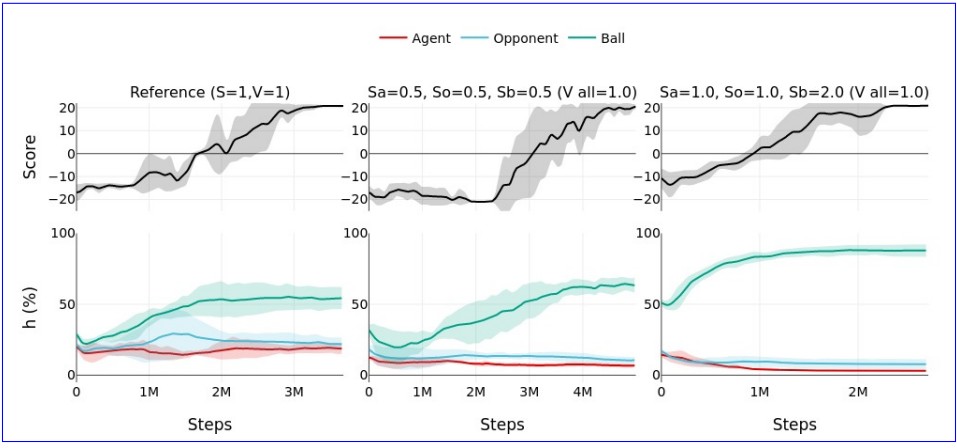

Figure 27: **Impact of object size on the attention profile.** Curves show reward progression (top) and h-profile trajectories (bottom) for A2C agents trained with different scaling factors for object size. The reference setting corresponds to $S = 1$ and $V = 1$. Scaling factors for the size of the agent, opponent, and ball are denoted by $S_a$, $S_o$, and $S_b$, respectively. Velocity scaling is kept constant ($V = 1.0$). The bold line represents the average over three agents, and shaded areas indicate one standard deviation. Attention profiles are computed using LRP.

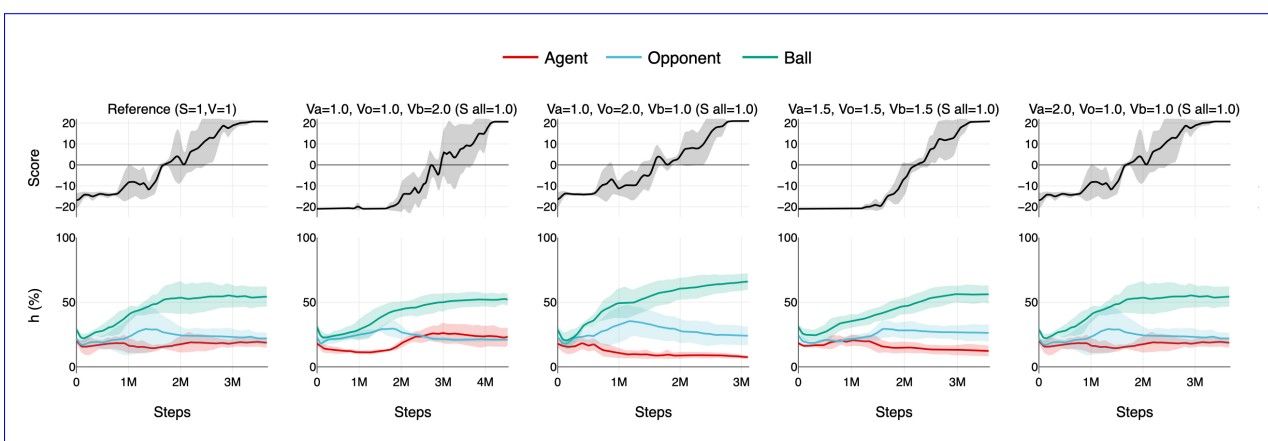

Figure 28: **Impact of object velocity on the attention profile.** Curves show reward progression (top) and h-profile trajectories (bottom) for A2C agents trained with different velocity scaling factors. The reference setting corresponds to $S = 1$ and $V = 1$. Scaling factors for the velocity of the agent, opponent, and ball are denoted by $V_a$, $V_o$, and $V_b$, respectively. Object sizes are kept constant ($S = 1.0$). The bold line represents the average over three agents, and shaded areas indicate one standard deviation. Attention profiles are computed using LRP.

