# OpenReview forum: "Attention Trajectories as a Diagnostic Axis for Deep Reinforcement Learning"
_TMLR — Under review for TMLR_

### Review · Reviewer_JFBG · 2025-12-27

**Summary Of Contributions:**

This paper proposes a quantitative, longitudinal method for interpreting DRL by aggregating saliency into hierarchical attention profiles over predefined objects/modalities and tracking these attention trajectories during training.
The method is demonstrated through three case studies of increasing complexity:
1. Atari: compares attention trajectories across four algorithms (A2C, PPO, DQN, QR-DQN) on four games, revealing algorithm-specific attention biases. DQN and QR-DQN develop high reliance on bricks in breakout and become vulnerable to visual perturbations of this features.
2. Custom Pong: Three variants with identical visuals but different reward functions demonstrate how reward design shapes attention allocation and can lead to unintended strategies.
3. Visuomotor Interactive Tasks: Extends the approach to multimodal inputs (vision  + proprioception) in muscle-actuated simulations, showing dynamic reallocation of attention across modalities during sequential task completion, and diagnosing potential overfitting to redundant sensory channels.

Strengths:
- Systematic quantitative approach: moves beyond qualitative, case-by-case saliency analysis toward reproducible, statistical comparisons of attention patterns.
- Each case study includes controlled experiments linking attention differences to observable behavior (perturbation robustness, ball preference, etc), providing causal evidence rather than just correlation.
- The authors show robustness of their findings by extending experiments to difference saliency methods, also provide open-source code for reproducibility, strengthening the reliability of the result.

Weakness:
- While the paper shows that attention profiles differ across algorithms, there is limited explanation of why (e.g. what properties of DQN/QR-DQN vs other algorithms cause different brick attention?)
- The method relies on predefined object/modality labels, make it harder to extend to unstructured inputs or relational feature like distances/velocities. Authors acknowledged this.

**Audience:**

Yes

**Audience Explanation:**

DRL interpretability remains a high-interest topic. The method is simple to apply, saliency-method agnostic, and the results are easily visualisable and interpretable. The authors also provide open-source code for easy application.

**Broader Impact Concerns:**

The work can improve safety and transparency by revealing brittle strategies early. The paper includes a brief impact statement which is appropriate.

**Claims And Evidence:**

Yes

**Claims Explanation:**

The paper presents coherent evidence of the observations and claims through case-studies. Behavioural tests and perturbation experiments tie attention differences to outcomes, and replication with alternative saliency methods strengthens the credibility. Statistical checks (ANOSIM, t-test) support group differences. Some analysis (e.g. sensitivity of h-profiles to label granularity, Fc selection_ could be expanded, but the main empirical claims are convincingly demonstrated.

**Requested Changes:**

- Quantify sensitivity to object granularity and labelling noise. Provide an experiment varying object definitions and show the effect on h-profiles. This can provide insights when labels are imperfect or less granular. (strengthening)
- Provide interpretation of algorithm-specific differences, can even be qualitative analysis. (strengthening)

---

### Review · Reviewer_4Ze1 · 2026-02-02

**Summary Of Contributions:**

In this paper the authors

1. Propose attention trajectories as a diagnostic tool in understanding DRL decision making.

2. Introduce h-profile that attends to various level of attention detail like object and modality level distributions.

3. Use the cases to demonstrate the effectiveness of using the attention trajectories.

**Additional Comments:**

NA

**Audience:**

Yes

**Audience Explanation:**

The paper claims salency maps are not effective and not reliable for a true understanding of the decision makings of DRL. The authors propose a novel way of using attention trajectories to make the decision making more transparent. Others in the TMLR audience can use the proposed idea in their DRL training, and obtain more interpretability, including indetification of bias.

**Broader Impact Concerns:**

No concerns. The paper is aimed at finding out algorithmic biases, unintended consequences of reward design, and overfitting. No impact concerns.

**Claims And Evidence:**

Yes

**Claims Explanation:**

Yes, the paper shows three useful cases where their proposed attention trajectories are useful and makes the decision making more interpretable.

**Requested Changes:**

1. The three cases chosen to analyze are very structured. It would also be interesting to see how h-profile behaves in unstructured and noisy RL environment. A discussion of the potential issues there can also be helpful. Though it is mentioned in the discussion section, an example of such a case would be helpful.

2. There are some naive baselines that can be compared against for sanity check. Examples - object size (pixel area), object motion magnitude, reward proximity. It would be interesting to add more simpler baselines to see the impact of the propose attention trajectory.

---

### Review · Reviewer_ykXp · 2026-05-31

**Summary Of Contributions:**

This paper proposes hierarchical attention profile (h-profile), which is a saliency map based post-hoc interpretability method to study RL agent behavior and learning dynamics. H-profile is computed by aggregating relevance scores over predefined objects via as subset of important last-layer neurons. Experiments are conducted across 4 RL algorithms on Atari, a custom Pong environment, and a bio-mechanical environment, providing novel insights into different algorithms' behavior and learning dynamics that are not available from analyzing reward alone.

**Audience:**

Yes

**Audience Explanation:**

I believe the paper is relevant to the TMLR audience because understanding and evaluating RL agents is a standing challenge. This paper proposes not only an effective method/metric but also an evaluation framework for doing so.

**Claims And Evidence:**

No

**Claims Explanation:**

This paper aims to demonstrate h-profile as a generalizable and effective method for analyzing RL agent behavior and learning dynamics. This is clearly demonstrated by the following:
* Showing that hierarchical object level aggregation can be applied to different saliency methods with generally consistent findings.
* Showing that tracking object level attention distribution provides novel insight to agent learning dynamics and failure modes (such as in DQN).
* The writing is clear and easy to understand.

I did raise below one question on whether the proposed method is applied to A2C and PPO's actor network vs DQN and QRDQN's value network which potentially casts doubt on the consistency of the results. I'm happy to revise my response to this question if this is clarified.

**Requested Changes:**

* This paper claims that h-profile can be combined with any saliency methods. This is support by additional results in the appendix. However, it would be great to highlight briefly in the main results section as this is a part of the key contribution.
* I might have missed this: for A2C and PPO, is the saliency method applied to the actor or critic? If the saliency method is applied to A2C and PPO's actors but DQN and QRDQN are critics/value networks, then it makes sense they pay attention to different objects and we can't really claim performance differences are due to attention differences? For example, value networks may use the number of bricks to predict how well the agent has performed which might not be strictly useful for actors.
* This paper obtained many interesting insights on agent behavior and learning dynamics from object attention distributions. However, these insights are arguably correlational rather than causal. I was wondering if it's possible to conduct interventional experiments to further validate these findings? For example, a major finding from case study 1 is DQN and QRDQN's suboptimal performance is correlated with paying attention to bricks. Is it possible to show that if one where able to make DQN pay less attention to bricks (e.g, by formulating that as a reward), then its performance can potentially improve?
* This paper proposes using saliency map to understand more nuanced behavior. Is it possible that even simpler metrics can do this job? For example, in the distractor pong environment, we can presumably measure just how often the agent hits or move towards ball 2. (Though I understand such metric development might be very cumbersome and less generalizable than the proposed method).

---

> ### Author Response · Authors · 2026-06-02
>
> Thank you very much for your constructive feedback and for highlighting the value of our evaluation framework. We clarify the points you raised below and outline the corresponding manuscript changes.
>
> **1. Actor vs. Critic Clarification**
>
> We interpret this comment as raising two related issues: whether h-profiles for actor-critic methods are computed from an actor or critic representation, and whether the LRP-specific neuron-filtering step introduces output-dependent differences.
>
> * The h-profile is computed from the same penultimate fully connected layer, denoted Fc for all four algorithms: A2C, PPO, DQN, and QR-DQN. In our A2C and PPO implementations, the actor and critic share the backbone network up to Fc, after which separate heads predict the policy and value. Thus, the Fc representation used for computing the h-profile is shared by both the actor and the critic. We chose this layer because it provides a comparable representational bottleneck across algorithms: although it is shaped by different learning objectives (policy/value losses for A2C and PPO, Q-value prediction for DQN, and distributional return prediction for QR-DQN), it sits immediately upstream of action selection and is therefore directly relevant to agent behavior. We agree that if A2C and PPO used entirely separate actor and critic networks, the claim that attention differences are indicative of behavioral differences would be more difficult to make. We will explicitly clarify this architectural detail in the revised manuscript.
>
> * For the LRP-based h-profile, the Fc-neuron selection step is output-dependent: relevance is propagated from the actor output for A2C/PPO, from Q(s,a) for DQN, and from Z(s,a) for QR-DQN. Crucially, the same qualitative differences, including increased attention to bricks in Breakout, are also observed with perturbation-based and gradient-based saliency, which compute h-profiles directly from Fc without this output-specific filtering. We will highlight this robustness result in the main text and refer to Appendix Figure 20.
>
> **2. Causal vs. Correlational Insights**
>
> Regarding the concern about correlation versus causation, we agree that trajectory analysis is primarily correlational. To provide stronger evidence, we conducted an additional intervention on the experience replay-buffer size in DQN and QR-DQN. Our hypothesis was that the increased attention to bricks in value-based methods was not primarily driven by the actor/value output distinction, but rather by differences in data exposure induced by experience replay. In particular, early rewarding transitions involving brick hits may be overrepresented in the replay buffer and disproportionately influence the learned representation. Consistent with this hypothesis, reducing the replay-buffer size reduced attention to bricks. For DQN, this reduction in brick attention was directly associated with improved robustness to brick perturbations. While this intervention manipulates the data distribution during training rather than attention itself, it provides interventional evidence that a mechanism affecting the distribution of states directly alters the learned h-profile and the resulting behavioral robustness. More details about this experiment can be found in our response to Reviewer JFBG.
>
> We will also clarify that we do not present attention as the causal mechanism driving behavior. Rather, h-profiles provide a diagnostic axis for studying how feature reliance develops during training which can be connected to behavioral strategies.
>
> **3. Generality over Simpler Metrics**
>
> We agree that task-specific measurements are highly useful. However, such metrics must be custom-designed for every individual environment, limiting their scalability. The objective of the h-profile is not to replace task-specific behavioral metrics, but to provide a general, object-level diagnostic framework. Because raw saliency maps alone have historically cast doubt on their own reliability and ability to reflect true network weights, we embed them in a broader framework that combines:
> (i) Attention quantification, (ii) longitudinal analysis, (iii) cross-condition comparison, (iv) behavioral grounding, (v) validation across multiple saliency methods.
> In this framework, behavior is used to ground and validate broader patterns of attention development and feature reliance that can be applied across diverse domains.
>
> **4. Multiple Saliency Methods in Main Text**
>
> Finally, we agree that the compatibility of the h-profile across multiple saliency methods is a major contribution that deserves higher visibility. We will revise the main results section to explicitly discuss this cross-method consistency. Additionally, we will introduce a summary figure illustrating the core components of our framework (quantification, longitudinal analysis, cross-condition comparison, behavioral grounding, and cross-method validation) and a clarification of what each component contributes to.

---

> > ### Comment · Reviewer_ykXp · 2026-06-21
> >
> > Thank the authors for the clarification. I have no further questions.

---

### Author Response · Authors · 2026-06-02

Dear reviewers and action editor,

We thank the reviewers for their constructive feedback. We are preparing a revised version of the manuscript incorporating the requested changes and will upload it in the coming days.

Best regards,

The authors